



# Controls on managed aquifer recharge through a heterogeneous vadose zone: hydrologic modeling at a site characterized with surface geophysics

Zach Perzan[1], Gordon Osterman[2,3], and Kate Maher[1]

[1]Department of Earth System Science, Stanford University, Stanford, California, USA
[2]Agricultural Research Service, U.S. Department of Agriculture, Davis, California, USA
[3]Department of Geophysics, Stanford University, Stanford, California, USA

**Correspondence:** Zach Perzan (zperzan@stanford.edu)

**Abstract.** In water-stressed regions of the world, managed aquifer recharge (MAR), the process of intentionally recharging depleted aquifers, is an essential tool for combating groundwater depletion. Many groundwater-dependant regions, including the Central Valley in California, USA, are underlain by deep vadose zones (ca. 10 to 40 meters thick), nested within complex valley-fill deposits that can hinder or facilitate recharge. Within the saturated zone, interconnected deposits of coarse-grained material (sands and gravel) can act as preferential recharge pathways, while fine-textured facies (silts and clays) accommodate

the majority of the long-term increase in aquifer storage. However, this relationship is more complex within the vadose zone. Coarse facies can act as capillary barriers that restrict flow and contrasts in matric potential draw water from coarse-grained flowpaths into fine-grained, low permeability zones.

To determine the impact of unsaturated zone stratigraphic heterogeneity on MAR effectiveness, we simulate recharge at a

Central Valley almond orchard surveyed with a towed transient electromagnetic system. First, we identified three outcomes of interest for MAR sites: infiltration rate at the surface, residence time of water in the root zone and saturated zone recharge efficiency, which is defined as the increase in saturated zone storage induced by MAR. Next, we developed a geostatistical approach for parameterizing a 3D variably saturated groundwater flow model using geophysical data. We use the resulting workflow to evaluate the three outcomes of interest and perform Monte Carlo simulations to quantify their uncertainty as

a function of model input parameters and spatial uncertainty. Model results show that coarse-grained facies accommodate rapid infiltration rates and that contiguous blocks of fine-grained sediments within the root zone are >20% likely to remain saturated longer than almond trees can tolerate. Simulations also reveal that capillary-driven flow draws recharge water into unsaturated, fine-grained sediments, limiting saturated zone recharge efficiency. Two years after inundation, fine-grained facies within the vadose zone retain an average of 37% of recharge water across all simulations, where it is inaccessible to either

plants or pumping wells. Global sensitivity analyses demonstrate that each outcome of interest is most sensitive to parameters that describe the fine facies, implying that future work to reduce MAR uncertainty should focus on characterizing fine-grained sediments.





# 1 Introduction

Groundwater overdraft has accelerated throughout the 21st century, threatening water supply for nearly 49% of the world's
population and 38% of irrigated agricultural land (Bierkens and Wada, 2019; United Nations, 2022). Managed aquifer recharge
(MAR), the process of intentionally recharging depleted aquifers, is an important tool for combating unsustainable groundwater
extraction, with the potential to offset up to 100% of overdraft in some groundwater basins (Alam et al., 2020). Of the many
types of MAR, flood managed aquifer recharge (flood-MAR) — in which individual sites (e.g., dedicated recharge basins,
agricultural land and restored floodplains) are inundated with excess water (e.g., storm water, snowmelt and reservoir release)
— holds large potential for expanding MAR implementation (He et al., 2021; Scanlon et al., 2016).

However, identifying and evaluating suitable flood-MAR sites presents numerous challenges. The topography, water con-
veyance infrastructure, surface water availability, groundwater quality, land use history, local water policy and, for agricultural
land, crop types must all be considered when evaluating MAR potential. Previous studies have combined several of these fac-
tors into multi-criteria decision analyses to identify MAR sites at the regional scale (e.g., Rahman et al., 2012; Russo et al.,
2015; O'Geen et al., 2015; Marwaha et al., 2021). Within California's Central Valley, O'Geen et al. (2015) developed the
Soil Agricultural Groundwater Banking Index, which uses a soil map, digital elevation model and map of crop types to rank
potential flood-MAR sites on agricultural land.

While these analyses provide a first-order constraint on site suitability, they only contain information on shallow soil layers
and cannot account for deeper geologic heterogeneity, which can impact the timing and extent of the increase in aquifer storage
induced by MAR (O'Leary et al., 2012; Maliva et al., 2015; Maples et al., 2019). Within the saturated zone, coarse-textured
sediments can transmit recharge water orders of magnitude faster than fine-grained material, leading to a rapid increase in
pressure in wells adjacent to MAR sites following inundation (O'Leary et al., 2012). Though fine-grained sediments are slow
to transport water, they can account for the majority of the long-term increase in saturated zone storage (Maples et al., 2019).

However, the impact of sediment texture on recharge becomes even more complex within the vadose zone. As in the saturated
zone, interconnected coarse-grained sediments can act as preferential recharge pathways during MAR, as revealed through
time-lapse geophysical imaging of flood-MAR (Sendrós et al., 2020b). By contrast, MAR modeling reveals that, in some
cases, these sediments actually inhibit flow as capillary barriers (Sallwey et al., 2018). In addition, even when coarse facies
accommodate preferential flow, contrasts in matric potential can draw water from high permeability, coarse-grained flowpaths
into low permeability zones, trapping up to 20% of water applied at the surface (Bahar et al., 2021; Qi and Zhan, 2022).
Neglecting these processes — and the lateral spreading they induce — can result in underestimating vadose zone travel times
by a factor of 2–3 (Fichtner et al., 2019).

Given the importance of quantifying geologic heterogeneity at recharge sites, near-surface geophysical methods have become
a popular tool for MAR site evaluation, as they can provide cost effective, high resolution imaging of the subsurface. Common
techniques include electrical resistivity tomography, transient electromagnetic, frequency domain electromagnetic, seismic
reflection and nuclear magnetic resonance methods (Haines et al., 2009; Maliva et al., 2009; Gottschalk et al., 2017; Goebel
and Knight, 2021; Sendrós et al., 2020a; Walsh et al., 2014; Parker et al., 2022). The use and availability of near-surface



geophysical data is expected to continue growing in the future. For example, airborne electromagnetic surveys have been used for groundwater exploration in Denmark (Møller et al., 2009; Siemon et al., 2009), Botswana (Sattel and Kgotlhang, 2004; Podgorski et al., 2013) and Australia Harrington et al. (2014). Within the United States, the California Department of Water
Resources recently completed airborne electromagnetic surveys of all critically overdrafted groundwater basins in the state.

Despite their growing popularity, however, geophysical methods have historically only been used to qualitatively interpret subsurface lithology. In one of the first quantitative assessments of recharge sites with geophysical data, Pepin et al. (2022) evaluated recharge pathways using a ray tracing approach at sites previously surveyed using a towed transient electromagnetic (tTEM) system. At each site, the authors developed 3D realizations of sediment type and measured the shortest path between
the surface and the water table along coarse-grained facies. This technique is computationally efficient, but it focuses on a single outcome of interest (recharge) and assumes water percolates exclusively through coarse sediments, which may not reflect reality.

When combined with geophysical data, hydrologic modeling could be used for more robust, quantitative assessment of recharge sites. Modeling aquifer response to recharge is not uncommon (Niswonger et al., 2017; Russo et al., 2015; Ringleb
et al., 2016; Alam et al., 2020), though few studies incorporate geologic heterogeneity into the model (e.g., Ganot et al., 2018; Maples et al., 2019). Using well logs and stochastic methods to approximate subsurface heterogeneity, Maples et al. (2019) simulated recharge at potential MAR sites in the northern Central Valley and showed that interconnected coarse-grained facies act as conduits for preferential flow. Through a 3-parameter global sensitivity analysis, the authors also showed that recharge is more sensitive to the average vertical hydraulic conductivity beneath a site than the hydraulic conductivity of either the coarse-
or fine-grained facies alone (Maples et al., 2020). However, Maples et al. (2019) focused on regional-scale MAR projects using a model with variations in heterogeneity over 100s of meters. The influence of meter-scale heterogeneity on recharge remains unclear, as even small blocks of fine-grained sediments can restrict flow.

In this work, to identify the impact of vadose zone heterogeneity on flood-MAR, we use a 3D variably saturated flow model to simulate recharge processes at a site previously surveyed with a tTEM system. We develop a workflow for parameterizing
a hydrologic model using high-resolution geophysical data (1-10 m pixel size), applying geostatistical methods to account for uncertainty in each step of the workflow. We then perform a suite of recharge simulations and quantify uncertainty on three distinct flood-MAR outcomes of interest. Global sensitivity analyses allow us to examine the extent to which the properties and distributions of fine-grained sediments control each outcome. They also determine the key subsurface properties that contribute to uncertainty in the simulation results, identifying the most important parameters to characterize when evaluating future MAR
sites.

## 2 Background

### 2.1 Recharge operations and outcomes

Inundation of a recharge site induces a complex response in the subsurface. During flood-MAR, temporary earthen levees are typically constructed around a parcel of land, which is flooded with surface water from a nearby river or canal (Fig. 1). Water





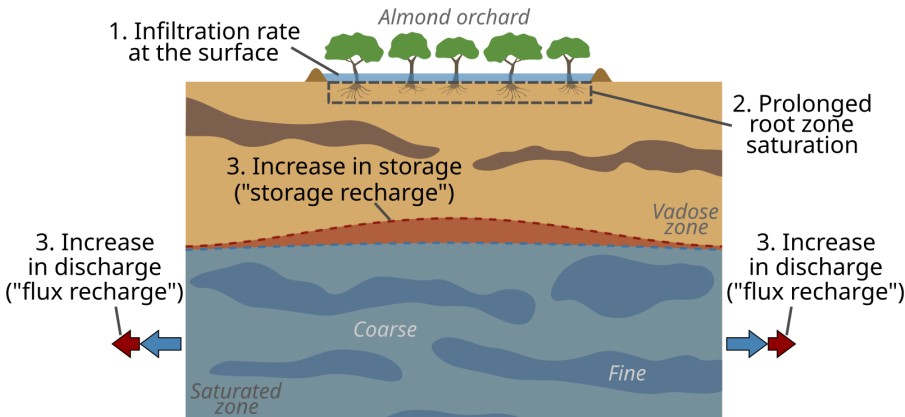

**Figure 1.** Theoretical cross-section of flood-MAR at an almond orchard, showing processes that determine the three outcomes of interest considered in this study: infiltration rate at the surface (1), root zone residence time (2) and saturated zone recharge efficiency (3). Saturated zone recharge efficiency consists of two components: storage recharge (the increase in saturated zone storage below a site) and flux recharge (the increase in lateral discharge away from the site below the water table). Red colors indicate long-term changes from pre-inundation conditions.

then infiltrates from the surface into the shallow soil and creates a pressure wave that moves through the system. Subsequent redistribution processes then transport water laterally and vertically under capillary forces and downward due to gravity. As the recharge pressure wave reaches the saturated zone, it forms a characteristic groundwater mound below the site (Bouwer, 2002). Mounding of the water table then causes an increase in lateral saturated zone flow away from the site, which increases storage in the broader aquifer (Fig. 1).

Along with these potential benefits, flood-MAR can have adverse effects. Prolonged saturation of the root zone can lead to soil anoxia and greenhouse gas production (Levintal et al., 2022). Recharge can also degrade groundwater quality by mobilizing geogenic contaminants (Fakhreddine et al., 2021) or flushing salts, fertilizers and pesticide byproducts from the vadose zone (Levintal et al., 2022).

Given these potential advantages and disadvantages of recharge, we quantify three primary outcomes of interest for flood-
MAR: infiltration rate at the surface, root zone residence time and saturated zone recharge efficiency. The infiltration rate across the surface — the rate of movement of ponded water into the shallow soil – can determine the success of a MAR project (Bouwer, 2002). Within California, high-magnitude flows used for flood-MAR typically only occur during short (<10 day) windows (Kocis and Dahlke, 2017). Sites with high infiltration rates can capture more water over these short windows, while also reducing the total area of land required for recharge. Moreover, maps of infiltration rate across a site are valuable for
identifying high infiltration zones; if water managers can isolate and inundate a portion of a site but still have high recharge efficiency, they decrease operating costs (i.e., constructing fewer earthen levees) and limit crop risk.





The second metric, root zone residence time, accounts for the duration of root zone saturation and risk of soil anoxia. In agricultural settings, soil anoxia can inhibit root growth, root respiration, shoot growth, seed germination, nutrient uptake and crop yield (Kozlowski, 1997). Soil anoxia also promotes the production of organic and inorganic compounds that are toxic to
some plants (Drew and Lynch, 1980). In addition, many tree crops have shallow root structures and can get blown over in high winds when the soil is saturated. Even at non-agricultural recharge sites, soil anoxia is undesirable as it can increase the release of greenhouse gases like carbon dioxide and nitrous oxide.

Finally, there are many potential metrics for quantifying the recharge efficiency of a MAR project, including the average infiltration rate across the surface (Heilweil et al., 2015), the infiltration rate minus the evapotranspiration rate (Dahlke et al.,
2018; Bachand et al., 2014), the fraction of applied water that reaches downgradient abstraction wells (Ganot et al., 2018) and the increase in hydraulic head across both the vadose and saturated zones (Russo et al., 2015; Maples et al., 2019). The primary goals of MAR — limiting water table decline, preventing land subsidence and increasing saturated zone storage — entail replenishing water (or increasing pressure) within the saturated zone (Morrell, 2014). Thus, we define recharge efficiency as the increase in saturated zone storage that results from MAR operations. Depending on the scale of the model, this includes the
increase in saturated zone storage below a site as well as the increase in lateral saturated zone flow away from the site (across model boundaries), which increases storage within the broader aquifer (Fig. 1). Note that this definition differs from previous work that equates recharge with deep percolation, which is the downward movement of water past the root zone. Though deep percolation water will likely reach the water table, it may take years to do so. This time lag impacts water management decisions and can vary from site to site, so we focus on saturated zone recharge. Due to a lack of water and sediment chemistry
data, we do not consider changes in groundwater quality during recharge. More details on each metric are described in section 3.4.1.

## 3 Methods

### 3.1 Study area

Our study site consists of an 800 m by 400 m almond orchard within the Tulare Irrigation District in the southern part of the
Central Valley (Fig. 2). The Tulare Irrigation District is underlain by a complex alluvial sedimentary system typical of the Central Valley, consisting of interconnected layers of sand and gravel embedded in a matrix of fine-grained sediments (Mid-Kaweah GSA, 2019). Regional studies have shown that the upper 30 m of the subsurface typically consists of interbedded deposits of gravelly sand, silty sand and silt, with a higher percentage of less permeable, clay-rich deposits below 30 m (Mid-Kaweah GSA, 2019). Depth to bedrock beneath the site is 350–400 m, though a low-permeability, laterally extensive aquitard,
the Corcoran Clay, underlies the site at a depth of approximately 150 m (Mid-Kaweah GSA, 2019). The Soil Agricultural Groundwater Banking Index (SAGBI) rates the site as "poor" (score 21/100; O'Geen et al., 2015), due to slow soil drainage at the site.

The local, semi-arid climate is characterized by dramatic deviations in precipitation from one year to the next (Dettinger et al., 2011). Between 1980 and 2021, total annual precipitation ranged from 6 cm (2013) to 42 cm (1998). Most annual





precipitation falls between November and March, which is the season in which Central Valley MAR operations typically occur. The Tulare Irrigation District operates several recharge projects in the area and has plans to expand in the future, with a particular focus on on-farm recharge (Mid-Kaweah GSA, 2019). However, no recharge operations have occurred at this site in the past.

## 3.2    Geophysical and geostatistical methods

### 3.2.1    Geophysical data acquisition

To characterize subsurface heterogeneity, the study site was surveyed with a towed transient electromagnetic (tTEM) system in fall 2017 (Behroozmand et al., 2019). The tTEM system can rapidly map meter-scale variations in subsurface resistivity over large areas. Given the link between resistivity and sediment texture (Knight and Endres, 2005), tTEM is a natural choice for recharge site assessment. In total, 46 line-km of tTEM data were acquired in the almond orchard and an adjacent field (Fig.

2). Within the orchard, tTEM data was collected in rows ∼6.8 m apart and, after data processing, resulted in a sounding every ∼10 m along each row. In the adjacent field, tTEM data was collected in lines 25–40 m apart with a sounding every 5 m along each line. These data were inverted in Aarhus Workbench (Aarhus Geosoftware) using a smooth, 1D spatially constrained approach (Auken et al., 2015). For each sounding, a global sensitivity threshold (Christiansen and Auken, 2012) was used to determine the depth of investigation (30–80 m), below which reliable estimates of resistivity cannot be recovered. The resulting

resistivity model (Fig. 3a) contains 24 layers. Layer thicknesses increase from 1 m at the surface to 9.7 m at the maximum depth of investigation (81 m). The resistivity model was used to estimate the depth to the water table at the site (45 m; Goebel and Knight, 2021), which is in agreement with nearby water table measurements (44 m).

### 3.2.2    Interpolation and extrapolation of geophysical data

Following geophysical inversion, the resistivity model consists of irregularly spaced point data. To transform this data onto a

rectilinear grid for use in a hydrologic model, we use sequential Gaussian simulation (SGSIM). SGSIM is a common technique for populating a grid with a Gaussian random variable without the smoothing effect of deterministic methods (e.g., ordinary Kriging). Because the hydrologic model domain extends beyond the bounds of the resistivity model, this procedure includes both interpolation (between any soundings >10 m apart) and extrapolation (vertically from the depth of investigation to 90 m and laterally up to 200 m away from the orchard boundaries). SGSIM accounts for the uncertainty in this process by generating

multiple, equally probable realizations of the variable of interest. The full workflow is shown in the supplementary material (Fig. S1) and described briefly here.

     First, resistivity values were transformed to a Gaussian distribution using a normal score transformation. An implicit assumption of SGSIM is stationarity, meaning that statistical properties of the random variable (e.g., the mean and variance) do not vary spatially throughout the domain. However, experimental variograms of the normal score values displayed a trend with

depth. To account for this, we decomposed each normal score value ($z(\mathbf{u})$, where $\mathbf{u}$ is a location in the domain) into trend ($m(\mathbf{u})$) and residual ($r(\mathbf{u})$) components (Wackernagel, 2003):





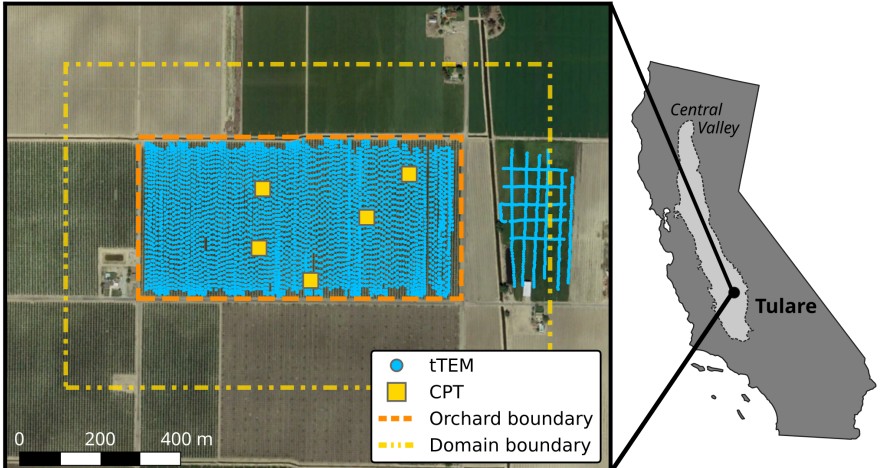

**Figure 2.** An aerial image of the study site and a map of California showing the location of the site within the Central Valley (light gray area). Individual tTEM soundings are shown as blue circles, while the location of each cone penetration test (CPT) is represented as a yellow square. The extent of the hydrologic model domain (yellow dashed line) is 200 m larger in every direction than the extent of the almond orchard (orange dashed line).

$$z(\mathbf{u}) = m(\mathbf{u}) + r(\mathbf{u}) \tag{1}$$

We modeled the trend using a radial basis function with a Gaussian kernel parameterized to capture the smooth, large-scale changes in normal score values. To extrapolate the trend laterally, for each geostatistical realization, 15 random 1D sounding profiles were copied from the tTEM survey to random locations outside the survey footprint. These 15 profiles are enough to ensure the extrapolated $m(\mathbf{u})$ values exhibit a similar trend beyond the orchard as within it. This is a logical assumption given that the characteristic trend within the study area is a decrease in resistivity with depth, which is consistent with regional surveys of resistivity (Knight et al., 2018) and sediment type (Mid-Kaweah GSA, 2019). Next, we transformed the residuals to a Gaussian distribution and generated 600 realizations of $z(\mathbf{u})$ on a 1200 x 800 x 90 m ($x$ x $y$ x $z$) uniform grid using SGSIM. This number of realizations was chosen so as to satisfy the approximate number of simulations required for global sensitivity analyses, described in section 3.4.2. For cell sizes in the $x$ and $y$ directions, we used the approximate spacing between tTEM soundings (10 m). For the $z$ direction, we used the finest vertical cell size in the inverted tTEM resistivity model (1 m). Following SGSIM, each realization of $z(\mathbf{u})$ was added to a unique realization of the extrapolated trend, $m(\mathbf{u})$. Finally, the combined trend and residual values were back transformed to resistivity units and down-sampled to the vertical resolution of the tTEM-derived resistivity model. An example realization is displayed in Fig. 3b.

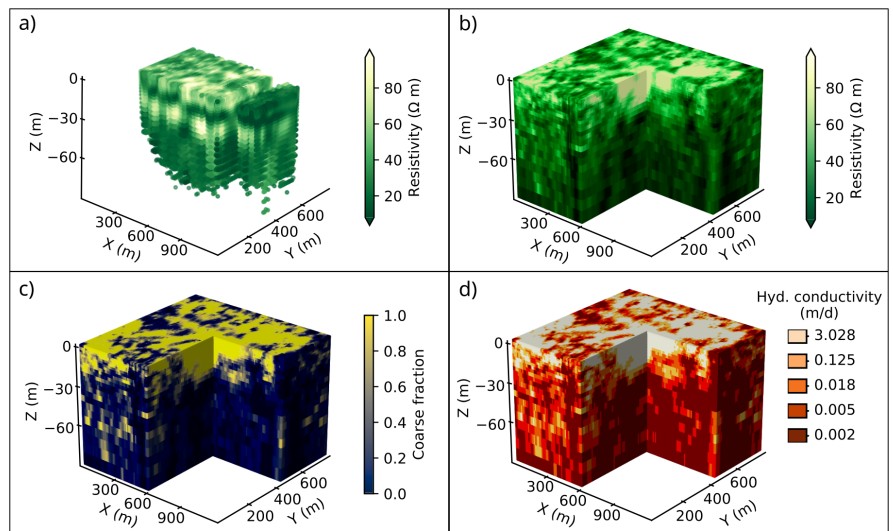

**Figure 3.** One example realization at each step of the workflow for parameterizing a hydrologic model with tTEM data. Inversion of the tTEM soundings produces a resistivity model of irregularly spaced points (a). The geostatistical workflow (section 3.2.2) then interpolates and extrapolates the resistivity model onto a rectilinear grid (b). A rock physics transform is then applied to create realizations of coarse fraction (c). Finally, after clustering each coarse fraction realization into distinct facies, each facies is assigned specific parameter values, such as horizontal hydraulic conductivity (d). All plots are shown with 10x vertical exaggeration.

### 3.2.3 Transforming resistivity to coarse fraction

To determine sediment properties from resistivity data, 1D tTEM resistivity profiles were compared with sediment types determined from collocated cone penetration test (CPT) logs. This procedure is described in detail in Goebel and Knight (2021) and briefly summarized below. Cone penetration tests, commonly used to determine sediment properties in unconsolidated

sediments (Lunne et al., 1997), are performed by slowly pushing a metal rod into the subsurface and recording resistance at the rod tip and along the rod sleeve. These two values are then used to estimate horizontal hydraulic conductivity (Robertson et al., 1992; Lunne et al., 1997) and determine the normalized soil behavior type — a classification of 9 discrete sediment types (Fig. S2a) — of a discrete depth interval (Robertson, 2009, 2016).

    At this site, CPT logs were acquired at 5 locations extending from the surface to depths of 19–32 m with a sampling interval

of 10 cm (Fig. 2). Logs of soil behavior type at these 5 locations were then compared to profiles of electrical resistivity recorded during CPT logging (Fig. S2a) and $k$-medoids clustering was used to classify soil behavior types with similar resistivity values (Goebel and Knight, 2021). This classification scheme resulted in a coarse-dominated cluster corresponding to high resistivity values ("clean sand to silty sand" and "gravelly sand to sand") and a fine-dominated cluster corresponding to low resistivity values (all other soil behavior types). Clustering results were then used to convert soil behavior type logs to binary coarse-fine

sediment logs (Fig. S2b).





The CPT data was then aggregated to the same resolution as the tTEM-derived resistivity model by calculating the fraction of coarse-classified sediment layers over a given depth interval, a statistic referred to as coarse fraction. For example, if a 4.8 m sequence of CPT log contains 1.2 m of "gravelly sand to gravel" (classified as coarse), 1.2 m of "clay" (fine) and 2.4 m of "sand" (coarse), then that cell would receive a coarse fraction value of 75% (Fig. S2c). Aggregating the CPT data resulted

in 62 depth intervals with paired coarse fraction and resistivity information. Though the logs do not extend to the full depth of investigation of the tTEM survey, the range of resistivity values in the overlapping profiles covered 97% of all resistivity values in the tTEM resistivity model (Fig. 3a). The eddy currents induced in the subsurface during a TEM survey are primarily oriented horizontally. If we assume that sediment layers within a single tTEM cell are also oriented horizontally, we can model the electrical current flow through each cell as current flow through a set of resistors in parallel. Algebraically, this is described

as:

$$\rho_{\text{tTEM}} = \left( C_F \frac{1}{\rho_{\text{coarse}}} + (1 - C_F) \frac{1}{\rho_{\text{fine}}} \right)^{-1} \tag{2}$$

where $\rho_{\text{tTEM}}$ is the resistivity of a tTEM cell, $C_F$ is the fraction of coarse layers within a cell, $\rho_{\text{coarse}}$ is the resistivity of the coarse-grained layers, and $\rho_{\text{fine}}$ is the resistivity of the fine-grained layers. Using the procedure of Knight et al. (2018), we then generate 1,000 bootstrapped samples of 62 paired $\rho_{\text{tTEM}}$ and $C_F$ values and use non-negative least squares to solve for $\rho_{\text{coarse}}$

and $\rho_{\text{fine}}$ for each sample. This technique produces a distribution of the expected resistivity values for coarse-dominated and fine-dominated end members (Fig. 4). The range of resistivity values for a given end member represents both uncertainty in the field data and the variability in resistivity that exists for a given sediment type due to other factors (water content, salinity, etc).

Typically, sediments exhibit different resistivity-lithology relationships above and below the water table (Knight et al., 2018; Knight and Endres, 2005). The procedure above was used to build resistivity-sediment type distributions for the vadose zone,

but could not be used for the saturated zone as none of the CPT logs extend below the water table. Instead, we examined comparable resistivity-sediment type distributions from airborne electromagnetic data over the site and nearby well logs (Knight et al., 2018). We then calculated the offset, in Ω·m, between the vadose zone and saturated zone distributions for a given sediment type. We calculated separate scaling factors for coarse- ($10^{-0.118}$ Ω·m) and fine-dominated ($10^{-0.110}$ Ω·m) sediments; the scaling factor for coarse sediments is expected to be of larger magnitude because they typically exhibit a larger difference

in saturation above and below the water table than do fine sediments. We then applied this scaling factor to the vadose zone CPT-derived resistivity-sediment type distribution to generate equivalent distributions for the saturated zone (Fig. 4a).

Next, we used the vadose and saturated zone resistivity-sediment type distributions to transform resistivity to coarse fraction values between 0 and 1. Using increments of 0.001, for each coarse fraction value in this range we calculated $\rho_{\text{tTEM}}$ 10,000 times using equation 2 with $\rho_{\text{coarse}}$ and $\rho_{\text{fine}}$ randomly sampled from the distributions in Fig. 4a. Paired coarse fraction and

$\rho_{\text{tTEM}}$ values (Fig. 4b) were then aggregated into bins by resistivity and used to calculate the distribution of coarse fraction values within each bin. Distributions for two of these bins (45 ± 0.1 Ω·m and 53 ± 0.1 Ω·m) are plotted in Fig. 4c. Finally, we used the binned distributions to transform each geostatistical realization of resistivity (section 3.2.2) to coarse fraction. For each geostatistical realization, we iterated through all resistivity bins and sampled from each bin's coarse fraction distribution. To



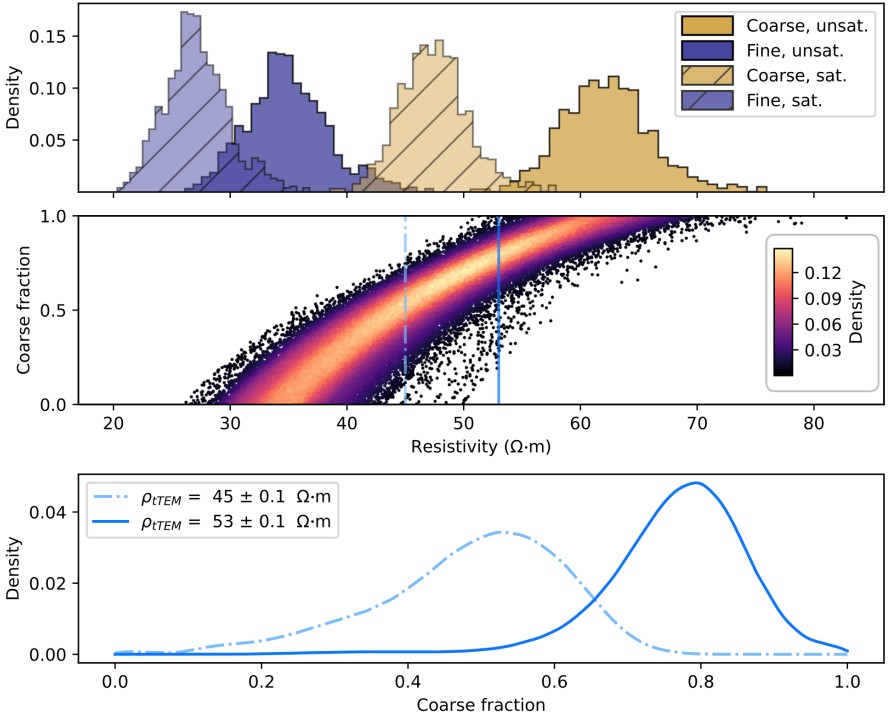

**Figure 4.** Resistivity-coarse fraction relationships used to create the rock physics transform. First, distributions of the expected resistivity values for coarse- and fine-grained end members (a) are generated from collocated cone penetration test (CPT) logs and tTEM soundings. We then sample from each distribution and use equation 2 to calculate paired coarse fraction and resistivity values (b). This process is performed separately for the unsaturated and saturated zones, though only the relationship for the unsaturated zone is plotted here. We then calculate the distribution of coarse fraction values in a series of discrete resistivity bins, two of which are plotted in (c).

vary the transform between each realization, we chose random quantiles at which to evaluate the coarse fraction distributions

for the lowest and highest resistivity values observed in that realization (*lowresq* and *highresq*, respectively). We then used linear interpolation between *lowresq* and *highresq* to determine quantiles at which to evaluate coarse fraction distributions for all resistivity bins between the two end points. This random sampling is advantageous over other methods (e.g., choosing the maximum likelihood value from each distribution) because it accounts for the uncertainty in the distribution of coarse fraction values corresponding to a single value of resistivity. The end result is a suite of 600 realizations of coarse fraction, several of

which are presented in Fig. 3c and Fig. S3.

### 3.3 Hydrologic model development

To model recharge at the site, we use ParFlow-CLM, which is a coupling of ParFlow, a hydrologic model, and the Common Land Model (CLM), a land surface model. ParFlow simultaneously solves the three-dimensional Richards' equation and the shallow water equations, while CLM uses a mass transfer approach to simulate water and energy fluxes from soil, snow



and leaf surfaces. For details on the governing equations of each model, readers are referred to previous work describing ParFlow (Ashby and Falgout, 1996; Jones and Woodward, 2001; Kollet and Maxwell, 2006), CLM (Dai et al., 2003) and the coupling between the two (Kollet and Maxwell, 2008; Maxwell and Miller, 2005). In this study, we use the kinematic wave approximation to simulate shallow overland flow (Kollet and Maxwell, 2006) and the van Genuchten formulation of the soil water retention curve (van Genuchten, 1980).

### 3.3.1 Boundary conditions and discretization

The hydrologic modeling domain measures 1200 m x 800 m x 150 m in the $x$, $y$ and $z$ directions, respectively, with uniform, 10 m cell sizes in the $x$ and $y$ directions. In the upper 90 m, cell thicknesses in the $z$ direction match the resolution of the tTEM cells (1–9.7 m). Below that depth, we add a single layer of 60-meter-thick cells to account for the estimated thickness of the unconfined aquifer below the tTEM depth of investigation (Mid-Kaweah GSA, 2019). In total, the domain contains 240,000 grid cells (120 x 80 x 25 cells in $x$, $y$ and $z$).

We impose a no flow boundary on the bottom of the domain, which approximates the depth to the Corcoran Clay, an aquitard. On the sides, we apply a constant head boundary equal to the estimated depth of the water table (45 m). During model development, we adjusted the lateral boundaries as far as 1000 m from the edge of the almond orchard and found that constant head boundaries 200 m away (Fig. 2) are sufficiently far to have minimal impact on model results (Table S1). The boundary conditions used at the top of the model vary depending on the model stage and are described below.


### 3.3.2 Model parameterization

The geophysical and geostatistical workflow above (sections 3.2.1 – 3.2.3) results in 600 unique realizations of one sediment property, coarse fraction, below the site. To estimate hydrologic properties of the subsurface (hydraulic conductivity, porosity, etc) from this metric, we assume that there are only two end member sediment types at the site — coarse and fine — and that each cell is a mixture of the two. We then define parameters for each end member and an averaging procedure to calculate properties of cells that are a mix of the two end members. This approach has been applied to Central Valley groundwater flow models in the past (e.g., Phillips and Belitz, 1991; Belitz and Phillips, 1995; Faunt, 2009) and is supported by the observation that sediment texture and hydraulic properties are spatially correlated in geologic media (Russo and Bouton, 1992). Nonetheless, this assumption is a simplification of the real groundwater system; its implications are discussed more fully in section 6.

Due to a paucity of site-specific data, we compile a range of potential parameter values and averaging schemes using literature data from the Central Valley (rows 1–12 in Table 1), with a preference towards published research in the vicinity of the site. For most parameters, we approximate prior uncertainty using a uniform or log uniform distribution, with lower and upper bounds as the minimum and maximum of the range of values reported in the literature. The one exception is horizontal 275 hydraulic conductivity, described below. Once each parameter's prior uncertainty has been defined, we use Latin hypercube sampling to sample from each distribution, jointly varying all parameters between every simulation. Latin hypercube sampling


**Table 1.** Parameter ranges sampled during Monte Carlo simulations

| # | Parameter[a] | Range | Distribution | Reference |
|---|---|---|---|---|
| 1 | Sat. hydraulic conductivity, coarse ($K_c$) | $10^{-0.3}$ to $10^{2.1}$ m/d | CPT data | b |
| 2 | Sat. hydraulic conductivity, fine ($K_f$) | $10^{-5}$ to $10^{-0.7}$ m/d | CPT data | b |
| 3 | Cond. power-mean exponent ($p_{Kz}$)[c] | -1 to 0 | Uniform | d |
| 4 | Porosity, coarse ($\phi_c$) | 0.22 to 0.40 m³/m³ | Uniform | e |
| 5 | Porosity, fine ($\phi_f$) | 0.23 to 0.47 m³/m³ | Uniform | e |
| 6 | van Genuchten $\alpha$, coarse ($\alpha_c$) | 0.5 to 7.8 (m⁻¹) | Log uniform[f] | e |
| 7 | van Genuchten $\alpha$, fine ($\alpha_f$) | 0.1 to 2.8 (m⁻¹) | Log uniform[f] | e |
| 8 | $\alpha$ power-mean exponent ($p_\alpha$) | 0 to 1 | Uniform | f |
| 9 | van Genuchten $n$, coarse ($n_c$) | 2.42 to 7.47 (-) | Log uniform[f] | e |
| 10 | van Genuchten $n$, fine ($n_f$) | 1.12 to 5.30 (-) | Log uniform[f] | e |
| 11 | Residual saturation, coarse ($\theta_{r,c}$) | 0.12 to 0.42 m³/m³ | Uniform | e |
| 12 | Residual saturation, fine ($\theta_{r,f}$) | 0.00 to 0.93 m³/m³ | Uniform | e |
| 13 | RPT quantile for lowest resistivity bin (*lowresq*) | 0.3 to 0.7 | Uniform | g |
| 14 | RPT quantile for highest resistivity bin (*highresq*) | 0.3 to 0.7 | Uniform | g |
| 15 | Inundation timing | {Once, monthly, semimonthly, or weekly} | Discrete | h |

*a*. Principal component scores from principal component analysis of all coarse fraction realizations are also included in global sensitivity analyses but are not shown in the table. *b*. Lunne et al. (1997), Robertson et al. (1992); *c*. Only used for averaging vertical conductivity. $p = 0$ indicates geometric mean and $p = -1$ indicates harmonic mean; *d*. Freeze and Cherry (1979), Fogg et al. (2000), Faunt (2009); *e*. Botros et al. (2009); *f*. Zhu and Mohanty (2002); *g* These are the quantiles used to evaluate the rock physics transform (RPT) for the lowest and highest bins of resistivity values. See section 3.2.3; *h*. Bachand et al. (2019), Ganot & Dahlke (2021), Murphy et al. (2021), Ma et al. (2022)

has some statistical advantages over random sampling (Saltelli, 2008) and ensures that all regions of the parameter space are sampled.

For hydraulic conductivity, we approximate prior uncertainty as the distribution of conductivity values calculated from CPT logs, with separate distributions for coarse- and fine-dominated sediments. Previous work has shown that cone penetration tests can effectively estimate hydraulic conductivity in alluvial sediments (Gribb et al., 1998; Voyiadjis and Song, 2003; Shen et al., 2015) and the range of values in each distribution agrees with previous conductivity estimates within the southern Central Valley (Botros et al., 2009; Faunt, 2009; Belitz and Phillips, 1995; Bertoldi et al., 1991; Johnson et al., 1968; Zamora, 2008). To calculate hydraulic conductivity in cells that are a mix of fine and coarse sediments, we assume a horizontally layered system and use the arithmetic mean for horizontal hydraulic conductivity (flow parallel to layers). A harmonic mean is typically used for flow perpendicular to layers (Freeze and Cherry, 1979), though the geometric mean may be more applicable in systems with interconnected coarse facies, like the Central Valley (Fogg et al., 2000; Faunt, 2009). Given this uncertainty, we use a power mean function to calculate vertical hydraulic conductivity and include the power mean exponent as an additional parameter during Monte Carlo simulations (rows 3 in Table 1), with values between -1 (harmonic mean) and 0 (geometric





mean). Because we use different horizontal and vertical hydraulic conductivity averaging schemes, conductivity is anisotropic in cells with coarse fraction between 0 and 1.

Parameter ranges for porosity, residual saturation, van Genuchten $\alpha$ and van Genuchten $n$ (van Genuchten, 1980) are derived from a database of 97 sediment cores collected on the east side of the southern Central Valley (Botros et al., 2009). We use the arithmetic mean when averaging porosity and residual saturation across sediment types. To average van Genuchten parameters,

we use the arithmetic mean for $n$ and a power mean averaging scheme for $\alpha$ (Zhu and Mohanty, 2002), with exponent values ranging from 0 (geometric mean) to 1 (arithmetic mean).

Before applying each parameter set to a realization of coarse fraction, we classify each domain into discrete hydrogeologic units. First, we bin coarse fraction values in each realization into 5 groups using $k$-means clustering. Cluster centers vary between each realization given that each realization contains a slightly different distribution of coarse fraction values. We

then iterate through each cluster and replace all coarse fraction values within that bin with the bin mean. We define a sixth hydrofacies for the bottom layer of model cells (90–150 m depth) and assume that coarse fraction in this facies is equal to the mean coarse fraction in the rest of the domain. Previous work modeling Central Valley recharge has relied on a similar facies designation for deep portions of an aquifer with limited geologic information (Maples et al., 2019).

### 3.3.3 Model stages

Recharge modeling at the site consists of three separate stages:

1. Spin up to steady state using constant forcing

2. Spin up for one year with meteorological forcing using CLM

3. Followed by both:

  (a) for flood-MAR simulations, inundation, then a relaxation period with meteorological forcing

(b) for reference simulations, two years of meteorological forcing without any inundation

To initialize pressure throughout the model domain, each simulation is spun up to steady state using a specified flux of 0.15 m/y applied at the surface. This value represents the net recharge rate (precipitation + irrigation - evapotranspiration) of a hypothetical almond orchard in the southern San Joaquin Valley, calculated using precipitation and reference evapotranspiration data from the California Irrigation Management Information System (CIMIS; http://www.cimis.water.ca.gov/) and localized

irrigation data from the California Department of Water Resources (DWR, 2019). Reference evapotranspiration was converted to crop evapotranspiration using monthly crop coefficients for almonds within the Central Valley (Doll and Shackel, 2015). Given the spatial variability of evapotranspiration and irrigation, we calculated separate recharge rates for 15 CIMIS stations between 1998 and 2015 (Fig. S4); the final rate used for model spin up (0.15 m/y) is a weighted average of these 15 individual rates, with stations weighted according to their proximity to the study site. We apply this forcing to each domain until the

annual change in subsurface storage is <0.01% (Ajami et al., 2014), which takes between 8 and 131 years depending on the simulation.





**Table 2.** Inundation scenarios used for flood-MAR

| Inundation frequency | Once | Monthly | Semimonthly | Weekly |
|---|---|---|---|---|
| Number of events | 1 | 4 | 8 | 16 |
| Recurrence interval (d) | — | 30 | 15 | 7 |
| Applied water per event (m) | 0.80 | 0.20 | 0.10 | 0.05 |
| Total applied water (m) | 0.80 | 0.80 | 0.80 | 0.80 |
| Avg. duration per event (hr) | 52 | 8.3 | 4.9 | 2.1 |

Next, each model is run for one year using meteorological forcing (Xia et al., 2012; Rodell et al., 2004) covering Feb 1, 2010 to Jan 31, 2011. We chose this time period so that inundation occurs in spring 2011, when there is surplus surface water available in the southern Central Valley for flood-MAR (Bachand et al., 2014). The timing of inundation is important, given that antecedent soil moisture may have an impact on recharge rates. During this step, we also simulate crop transpiration and irrigation using CLM. Irrigation is applied between 7 AM and 5 PM from March through October with daily irrigation rates set to 105% of daily crop evapotranspiration (Doll and Shackel, 2015).

The pressure distribution at the end of the CLM spinup period is then used as the initial condition for two separate simulations, run concurrently: a flood-MAR simulation and a reference simulation without MAR. Among Central Valley almond orchards, on-farm recharge varies both in the amount of water applied during a season (0.065 – 0.76 m) and in the frequency of recharge events, ranging from weekly to once per season (Ganot and Dahlke, 2021; Bachand et al., 2014; Murphy et al., 2021). To investigate the effects of flood frequency on each outcome of interest, we vary the frequency of inundation as an additional parameter (Table 1). In all cases, a total of 256,000 $m^3$ (0.8 m/$m^2$) of water is applied to the orchard between February and May 2011, though we vary the number and frequency of events across 4 scenarios (Table 2).

To implement each recharge event in ParFlow, we apply a specified head at the upper boundary to cells within the almond orchard and a no flow boundary condition to the cells outside it. At the start of an inundation step, the specified head over the orchard is equal to the total amount of water applied for that event (e.g., 0.20 m for monthly inundation). For each subsequent time step, the specified head is decreased by the amount of water that infiltrated below ground in the previous time step. This loop is then repeated until no water is left on the surface. This scheme is a computationally efficient approximation of a falling head infiltration that allows for heterogeneous infiltration with ponded water that flows laterally towards preferential flow paths. Water applied to the orchard cannot move laterally over the surface outside the bounds of the orchard, which mimics an on-farm recharge operation that use check dams to isolate the recharge area.

Between subsequent recharge events and following the final recharge event, we implement a relaxation phase in which meteorological and irrigation forcing is applied at the surface using CLM. Each flood-MAR simulation consists of a single season of recharge in spring 2011 followed by a relaxation period until Feb 1, 2013, two years after initial inundation. To investigate changes in recharge efficiency beyond 2 years, 10 randomly selected simulations were run for an additional 8 years (ending on Feb 1, 2021) using meteorological and irrigation forcing without any additional recharge events.



Reference simulations cover the same time period as flood-MAR simulations, but do not include any inundation. Meteorological and irrigation forcing is applied for the entire two-year period using CLM. Comparing reference simulations to

simulations with flood-MAR permits differentiating the effects of recharge from other model processes that occur naturally over the same time period (e.g., an increase in vadose zone storage due to above average precipitation).

## 3.4    Model output analysis

### 3.4.1    Evaluating recharge outcomes

As discussed in section 2.1, for each simulation, we evaluate three recharge outcomes of interest:

1. Infiltration rate at the surface: To create maps of infiltration rate across the almond orchard, for each simulation we compute the time-varying volumetric flux of water across the surface into the domain. This is a spatially resolved flux, with variable infiltration rates from one surface cell to the next. We then tabulate the cumulative volume of water that flowed into each individual surface cell across all recharge events. To more easily compare infiltration maps with ponded depth, we divide cumulative volume by surface area and represent the infiltrated volume as a length.

2. Root zone residence time: To evaluate the duration of root zone saturation, we analyze the time series of water content within the root zone for each simulation. Although maximum rooting depth of almond trees can extend to 2 m, effective rooting depth typically ranges from 0.3–1.0 m (Andreu et al., 1997; Koumanov et al., 2006), so we define the root zone as the upper 1 m of the subsurface. Depending on the variety of rootstock, almonds can tolerate 2–14 days of saturation before bud break (December–early February) and 48 hours after bud break (late February–November) without decreased

plant growth or crop yield (O'Geen et al., 2015). In our simulations, most recharge events occur after bud break, so we assume that trees can only withstand 48 hours of saturation. For each root zone cell, we then calculate the percentage of simulations in which the cell exceeded 48 hours of continuous saturation over the 2 year inundation-relaxation cycle. Note that this 48-hour threshold does not directly account for the risk of tree mortality during high wind events — which can occur even during brief periods of saturation — though simulations that exceed the threshold have higher risk than

those that do not.

3. Saturated zone recharge efficiency: We define the final outcome of interest, recharge efficiency, as the increase in saturated zone storage below a site plus the increase in lateral saturated zone flow away from the site, which increases storage in the broader aquifer. The two components of recharge are conceptualized in Fig. 1 and contribute to the recharge volume, $R_{vol}$, at time $t$ as:

$$R_{vol}(t) = \sum_{i=1}^{t}(Q_{SZ,i} - Q_{SZ,0})\Delta t + S_{SZ,i} - S_{SZ,0} \tag{3}$$

where $Q_{SZ,0}$ is the saturated zone flux out of the domain prior to MAR, $Q_{SZ,i}$ is the lateral saturated zone flux at time $t$, $\Delta t$ is the time step, $S_{SZ,0}$ is total saturated zone storage prior to recharge and $S_{SZ,t}$ is the saturated zone storage at





time $t$. Thus, flux recharge $(Q_{SZ,i} - Q_{SZ,0})$ is cumulative over time, while storage recharge $(S_{SZ,t} - S_{SZ,0})$ is transient. Recharge volume varies over time due to both anthropogenic and natural processes. To calculate the portion of $R_{vol}$ that is due to MAR, we calculate the difference in recharge volume between reference and flood-MAR simulations, then normalize it by the volume of water applied at the surface ($V_{app}$):

$$R_{eff}(t) = \frac{R_{vol,t}^{MAR} - R_{vol,t}^{ref}}{V_{app}} \times 100 \qquad (4)$$

where $R_{eff}(t)$ is the saturated zone recharge efficiency at time $t$, $R_{vol,t}^{MAR}$ is the recharge volume at time $t$ for a flood-MAR simulation and $R_{vol,t}^{ref}$ is the recharge volume at time $t$ for the corresponding reference simulation.

To facilitate comparison with previous work, we also quantify deep percolation induced by MAR. Deep percolation is calculated using a modified version of equation 3 in which we replace saturated zone storage ($S_{SZ}$) with storage below the root zone (below 2 m) and saturated zone flux ($Q_{SZ}$) with lateral flux out of the domain below the root zone. As with saturated zone recharge efficiency, we then normalize this volume by the volume of water applied at the surface and express it as a percentage.

### 3.4.2 Global sensitivity analyses

To identify key parameters and processes affecting recharge, we perform a global sensitivity analysis for each outcome of interest. We utilize distance-based generalized sensitivity analysis (DGSA; Fenwick et al., 2014; Park et al., 2016), a computationally efficient technique that has been successfully applied to groundwater flow and transport models in the past (Hermans et al., 2018; Hoffmann et al., 2019; Perzan et al., 2021). We chose this form of sensitivity analysis because DGSA can take discrete parameters as input (e.g., inundation frequency) and can analyze high dimensional model responses (e.g., an infiltration map), but only requires a limited number of model simulations compared to other techniques, like variance-based methods.

In DGSA, simulations are clustered into groups with similar model responses (e.g., similar recharge efficiency) using the pairwise distance between responses. The distribution of an input parameter (e.g., porosity of the fine-grained end member) is then compared between each cluster; if the prior distribution of a parameter is significantly different from the parameter's distribution within a cluster, then the model is said to be sensitive to that parameter. The area between empirical cumulative distribution functions (CDFs), referred to as the CDF distance, is used to quantify the degree of similarity between parameter distributions. Large CDF distances could result either from model sensitivity or from random model variability due to the limited sample size, so a bootstrapping procedure is used to test the statistical significance of observed CDF distances. If the observed distance is larger than the bootstrapped distance at a chosen significance level, then that parameter is said to be influential (sensitivity metric > 1). If the observed distance is smaller than the bootstrapped distance, the parameter is insignificant (sensitivity metric < 1).

In the present study, we cluster simulations on the pairwise Euclidean distance between model responses with *k*-medoids clustering, using a different distance for each outcome of interest. For saturated zone recharge efficiency, we compute distances directly on each simulation's $R_{eff}$ value at two years. For infiltration and root zone residence time, we first reduce the



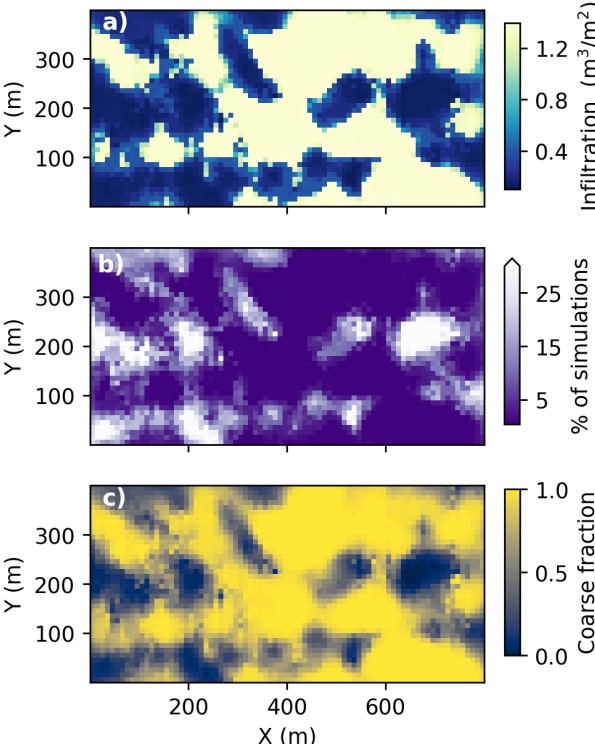

**Figure 5.** Maps of the first two outcomes of interest, aggregated across all 284 simulations. All simulations received 0.8 m of inundation, though the median volume of water that infiltrated into each surface cell (a) ranges from 0.10 to 1.40 m$^3$/m$^2$. The percent of simulations in which each surface cell exceeded the 48-hour root zone saturation threshold is shown in (b), with the median coarse fraction of each surface cell shown in (c). Note that each map only shows the portion of the model domain immediately below the almond orchard.

dimension of each model response (80- x 40-pixel images) using a form of principal component analysis known as eigenimage
decomposition. We then compute the distance between each simulation's eigenimage scores, using the minimum number of scores required to explain 98% of variance (typically, 40-50 components). Finally, we cluster simulations using these distances and perform the bootstrapping procedure described above to calculate sensitivity metrics.

In each sensitivity analysis, we include all model input parameters listed in Table 1. For a given simulation, a change in model response can result either from explicit perturbation of these input parameters or from the particular spatial configu-
ration of hydrofacies for that simulation. To account for this spatial uncertainty in sensitivity analyses, we perform principal component analysis across all realizations of coarse fraction (Fig. 3c) and include each simulation's principal component scores as additional input parameters to DGSA.





## 4 Results

### 4.1 Coarse fraction models

The geostatistical workflow described in sections 3.2.1–3.2.3 produced 600 realizations of coarse fraction across the model domain, four of which are shown in Fig. S3. The upper portion of the domain is dominated by coarse materials (mean coarse fraction 0.52 from 0 – 14 m depth) with a large proportion of fine-grained sediments at depth (mean coarse fraction of 0.14 from 28 – 90 m depth). However, pockets of fine-grained sediments as large as 100 m diameter occur throughout the domain, even within the coarse-dominated shallow vadose zone. Each realization displays similar coarse fraction values immediately

below the orchard (standard deviation ~0.04) with increasing variance towards the domain boundaries (standard deviation ~0.40; Fig. S5). Within the tTEM survey footprint, variance is highest at the transition zones between 100% coarse and 100% fine-grained sediments (standard deviation 0.2; Fig. S5). Finally, even though tTEM data were acquired at lower density within the adjacent field than within the almond orchard (Fig. 2), uncertainty on coarse fraction is similarly low within both regions (Fig. S5).

### 4.2 Outcomes of interest


In total, we performed 600 unique model simulations. Of these, 316 (53%) failed to complete all modeling stages, either due to invalid parameter combinations or because the ParFlow solvers failed to converge. Approximately half of the failed simulations did not complete initial model spin up (stage 1 in section 3.3.3) because permeability was too low to accommodate the specified flux across the surface, leading to a buildup of pressure in the top layer of model cells. Given that this specified

flux was calculated from observed data, we presume that these failed simulations reflect implausible parameter combinations for this site. Outcomes of interest calculated from the remaining, successful simulations are discussed below.

### 4.2.1 Surface infiltration rate

Across all simulations, the average infiltration rate during flood-MAR is $0.15 \pm 0.29$ m/hr. Infiltration rates are typically higher for simulations with smaller, more frequent inundation events: a mean of 0.06 m/hr for simulations with a single 80-cm

recharge event as compared to a mean of 0.24 m/hr for simulations with weekly 5-cm recharge events. Within each recharge event, infiltration rates slow down over time; among simulations with a single 80-cm recharge event, on average, the infiltration rate during the last time step is 17% of the initial rate.

Infiltration rates also vary with sediment type; on average, cells with coarse fraction >0.6 account for 91% of the volume of water that infiltrates into the subsurface even though they only make up 66.9% of orchard surface area. Maps of infiltration

below the orchard (Fig. 5a and Fig. S6) show that water preferentially infiltrates through coarse-grained cells, with 1.8–458 (median 10.7) times more water infiltrating into 100% coarse cells than into 100% fine-grained cells. Eigenimage decomposition and $k$-medoids clustering of all infiltration maps reveals two distinct groups of simulations. Simulations in both clusters





exhibit preferential infiltration through coarse facies, though the extent of preferential infiltration is greater in one cluster than the other (Fig. S6).

For a small subset of simulations (1.8%), the initial infiltration rate through fine-grained facies is faster than through coarse-grained facies. However, this relationship quickly switches once sediments become saturated (<1 hour), implying that coarse facies briefly acted as capillary barriers. Within each simulation, the greatest infiltration rates typically occur within coarse-grained cells on the edge of the orchard. These cells can accommodate more infiltration because — once below the surface — water can spread laterally into adjacent, dry cells beyond the orchard boundary.

### 4.2.2    Root zone saturation

In 65% of simulations, at least one root zone model cell remains saturated longer than 48 hours, the estimated time almond trees can endure soil anoxia without diminished growth or yield. Of the cells that exceed this threshold, 81% are predominantly fine-grained, with coarse fraction <0.6 (Fig. S7). Conversely, in 8% of simulations (23/284), at least one 100% coarse-grained cell exceeds the saturation threshold. Aggregating results across simulations reveals a similar trend; contiguous blocks of fine-460    grained sediments are >20% likely to remain saturated for at least 2 days following inundation (Fig. 5). Collectively, this area accounts for 9% of the orchard or 2.85 hectares (7.0 acres).

The increase in root zone saturation also drives a minor increase in transpiration within the almond orchard. In the first week after inundation, transpiration is an average of 37% higher in flood-MAR simulations than in reference simulations (2.16 vs 1.57 mm/d), which is reflective of the early growing season in California's Mediterranean climate. After 7 days, the difference 465    decreases and mean transpiration for the remainder of the growing season is only slightly higher (2.2% or 0.09 mm/d). Over the whole simulation period, increased transpiration only accounts for ∼4.8% (0.038 m) of the total applied water.

### 4.2.3    Saturated zone recharge efficiency

Inundation at the surface produces a pressure pulse that cascades through the subsurface, resulting in a transient increase in saturation throughout the vadose zone (Fig. 6 and S8). The pulse moves rapidly past the root zone towards the water table. It is 470    important to note, however, that this rapid change in storage corresponds to the transmission of pressure and not to the direct movement of water particles. Water applied at the surface displaces water in the root zone, which displaces water below the root zone, and so on. Thus, though saturation may increase at 30 m depth, it does not necessarily indicate that water particles applied at the surface have moved to this depth.

As the pressure pulse moves deeper, it causes the water table to rise (Fig. 6), which represents an increase in saturated 475    zone storage (storage recharge; Fig. 1). Mounding of the water table induces an increase in discharge out the sides of the domain, which constitutes recharge to the broader aquifer system (flux recharge). Temporal changes in these two components of recharge are plotted for a single simulation in Fig. 7. Storage recharge is the first component to increase following inundation, starting to rise between 34 and 683 days (mean 214 days) after the first inundation event. As storage recharge peaks and slowly decreases, flux recharge decreases as well. However, since flux recharge is cumulative over time (equation 3), overall saturated 480    zone recharge efficiency continues to increase until the end of the simulation (Fig. 7).

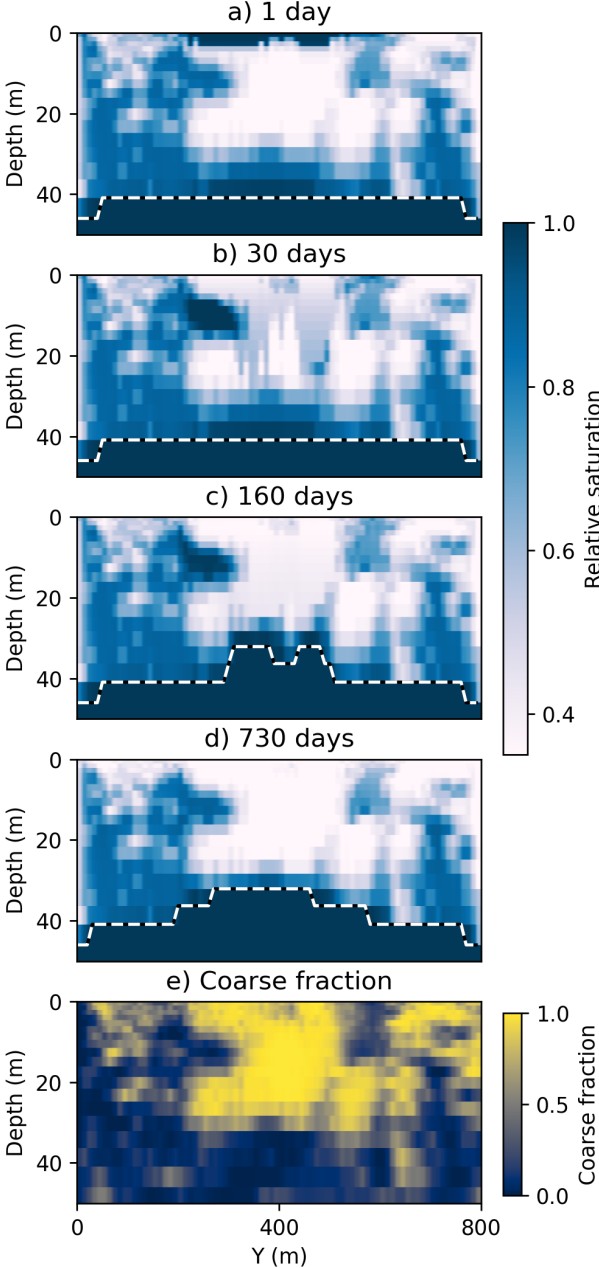

**Figure 6.** Vertical cross-sections of relative saturation (a-d) and coarse fraction (e) for a single simulation for 2 years (730 days) following inundation. Transient increases in saturation within coarse-grained sediments can be observed at 30 days and within fine-grained sediments at 30 and 60 days. The white dashed line shows the position of the water table, with obvious water table mounding at 160 and 730 days. Only the upper 50 m of the model domain is shown.



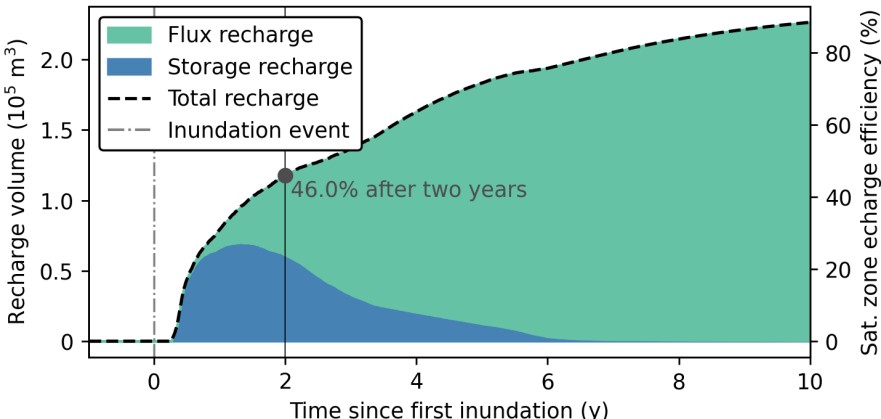

**Figure 7.** Temporal changes in the two components of saturated zone recharge efficiency — storage recharge (blue) and flux recharge (green) — for a single simulation for 10 years following inundation. Flux recharge is cumulative over time, while storage recharge is transient. The recharge efficiency of this simulation is 46% after two years, which is higher than the median across all simulations (32.7%).

Among all simulations, deep percolation increases rapidly following inundation. Fourteen days after the final inundation event, the increase in deep percolation accounts for $88.3 \pm 17\%$ of the volume of water applied at the surface (Fig. 8a). By comparison, saturated zone recharge efficiency is $32.7 \pm 26.3\%$ after two years (Fig. 8b). Ten of these simulations were run for an additional 8 years (total of 10 years post-inundation) and results show that saturated zone recharge efficiency continues to increase beyond 2 years (Fig. 7). However, none of the simulations reach 100% recharge and, though some get close, they plateau around 95% (e.g., Fig. S9). Given this paper's focus on short-term recharge outcomes (<2 years), more long-term simulations are needed to confirm that these results are consistent across the full parameter space.

Coarse- and fine-grained facies respond differently to recharge, especially between the bottom of the root zone (2 m) and the water table (45 m). Coarse facies in this zone respond rapidly to inundation; on average, water content begins increasing 3.5 days after the first recharge event and peaks 11.2 days after the last event (Fig. 9a). In contrast, water content in the fines begins increasing after 22.2 days and peaks 246 days after the final inundation event. Fine-grained sediments are also slow to drain following inundation; two years after the first recharge event, storage in fine facies between 2 and 45 m depth is on average 94,000 m³ higher than in reference simulations, which represents 37% of the volume of water applied at the surface (Fig. 9b). By comparison, coarse facies hold significantly less water than the fines after two years ($P <0.001$, paired $t$-test), a median increase of 10,700 m³ (4.2% of applied water) relative to reference simulations.

**4.3 Global sensitivity analyses**

Global sensitivity analysis results are presented in Fig. 10. For each outcome of interest, the most influential parameter is the hydraulic conductivity of the fine facies. Saturated zone recharge efficiency is also sensitive to the porosity and residual saturation of the fines, while root zone residence time is sensitive to residual saturation of the fines and the power mean exponent



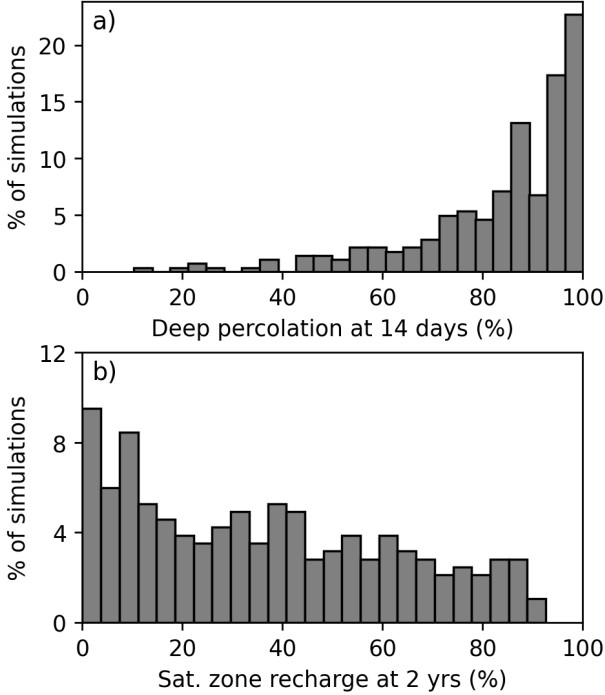

**Figure 8.** Histogram of deep percolation efficiency at 14 days (a) and saturated zone recharge efficiency at 2 years (b) for all 284 simulations. Deep percolation refers to the proportion of water applied at the surface that has moved beyond the root zone within 14 days of the final inundation event. Saturated zone recharge efficiency ranges from 0.28% to 87.7% (median 32.7%), though the distribution is skewed towards lower values.

500    used to average vertical hydraulic conductivity in cells with coarse fraction between 0 and 1 (section 3.3.2). Infiltration rate is the only outcome of interest sensitive to parameters that explicitly describe coarse facies; both the hydraulic conductivity and porosity of coarse-grained sediments are influential. Other parameters, like inundation frequency and the quantiles used for geophysical inversion (*lowresq* and *highresq*), are not influential. Similarly, none of the outcomes of interest are sensitive to metrics describing spatial uncertainty (mean coarse fraction, variance in coarse fraction, or coarse fraction principal component

505    scores), implying that the subsurface heterogeneity at this site is well characterized via tTEM. Finally, the sensitivity metrics of certain parameters were statistically inconclusive (white bars in Fig. 10). While these parameters may impact the outcomes of interest, they are not as influential as the critical parameters discussed above (red bars in Fig. 10).

## 5   Discussion

Managed aquifer recharge is an essential management tool for combating groundwater depletion within overdrafted groundwa-

510    ter basins (Alam et al., 2020), but successful implementation requires selecting a suitable recharge site. The results presented



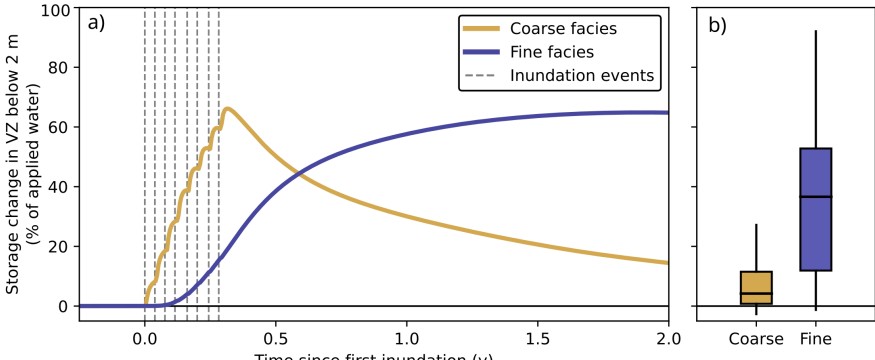

**Figure 9.** Storage change in the vadose zone below 2 m (a) for a simulation with low saturated zone recharge efficiency (8.7% after two years). The percent change in vadose zone storage at two years for all 284 simulations is shown as a box plot in (b). Paired *t*-test results indicate that the difference in the two populations in (b) is statistically significant ($P < 0.001$). In both subplots, coarse facies (yellow) are defined as any model cells with coarse fraction $\geq 0.5$; fine facies (blue) correspond to coarse fraction $<0.5$.

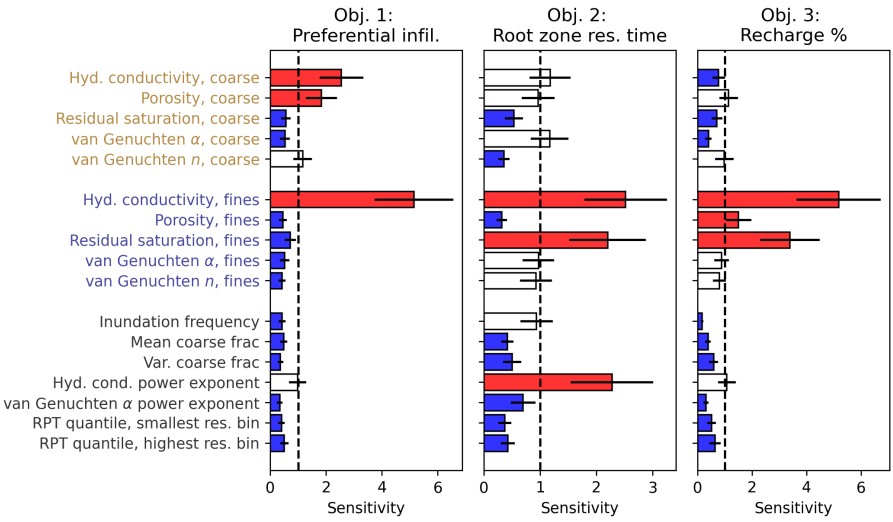

**Figure 10.** Distance-based global sensitivity analysis (DGSA) results for three outcomes of interest. Parameters are grouped according to whether they describe coarse facies (yellow labels), describe fine facies (blue labels) or are universal (black labels). Bars are colored according to whether a parameter is influential (red, sensitivity metric > 1), insignificant (blue, sensitivity metric < 1) or in between (white, sensitivity metric confidence interval includes 1). The principal component scores on coarse fraction for each simulation were included as additional parameters in DGSA, but are not shown here as none are influential.

here demonstrate a workflow for performing detailed recharge site evaluation using geophysical data and hydrologic modeling. Results also highlight key subsurface processes that can either inhibit or facilitate recharge. Previous work has shown that




subsurface heterogeneity often determines the efficiency of a recharge site (Maples et al., 2019). Many of the results herein support these findings, while furthering the discussion by identifying additional important processes that can limit recharge.

## 5.1 Outcomes of interest

As expected, maps of infiltration rate and root zone residence time align closely with maps of sediment texture, with coarser sediments yielding higher infiltration rates and shorter periods of saturation. Average infiltration rates are comparable to those measured during flood-MAR in the Central Valley (Bachand et al., 2014; Ganot and Dahlke, 2021) and are within the range of values suitable for surface spreading sites (Bouwer, 2002), implying that infiltration rate is unlikely to be a limiting factor at this site. Infiltration rate results also suggest that coarse facies briefly acted as capillary barriers in a small fraction of simulations (1.8%), which supports previous work showing that capillary barriers can form during MAR, but only under specific parameter combinations (Sallwey et al., 2018).

Root zone saturation risk, on the other hand, may discourage flood-MAR implementation, given that 9% of the orchard is >20% likely to remain saturated long enough to decrease crop yields. Given the high annual revenue ($3800–$7900 US-D/acre/year) and narrow profit margins of most almond orchards (USDA, 2022), even a small amount of crop risk represents a large financial risk to growers. Quantifying this risk (e.g., Fig. 5) allows water managers to appropriately compensate or incentivize farmers for implementing flood-MAR. Moreover, growers can use risk maps to develop an implementation strategy, enabling them to compare the cost of constructing additional earthen levees to isolate inundation to low-risk zones with the corresponding decrease in crop risk.

Importantly, the correlation between sediment type and both infiltration rate and root zone saturation implies that these two outcomes of interest can be approximated directly from coarse fraction data, as long as precise estimates are not required. In our simulations, cells with coarse fraction <0.6 are 8.7 times more likely to experience extended root zone saturation and have infiltration rates 4.7 times lower than cells with coarse fraction >0.6. Given these statistics, this coarse fraction threshold could be used to identify areas with high infiltration rates and low saturation risk, limiting the need for hydrologic modeling. It is important, however, to select a threshold coarse fraction rather than a threshold resistivity value because resistivity-lithology relationships differ from site to site (Knight et al., 2018).

While the relationship between sediment texture and infiltration rate is relatively straightforward, the influence of sediment texture on saturated zone recharge efficiency is more nuanced. In the short term, the recharge-induced pressure pulse moves most rapidly through coarse facies, which agrees with previous work (e.g., Maples et al., 2019). However, over the long term, fine-grained sediments absorb a large fraction of recharge water, especially within the vadose zone. This trapping of water limits saturated zone recharge efficiency, as fine facies between the bottom of the root zone and the water table can take years to drain (Fig. 9). In the meantime, water within these sediments is inaccessible to both plants and pumping wells. In addition, though the pressure pulse initially moves downward through coarse facies, the subsequent redistribution of pressure occurs in three dimensions. This implies that, even if inundation is restricted to coarse-grained sediments, adjacent fine facies may limit recharge by drawing pressure (and water) laterally away from interconnected, coarse-grained flowpaths.



## 5.2 Global sensitivity analyses

Global sensitivity analysis results further highlight the importance of capillary trapping of water in fine-grained sediments. Saturated zone recharge efficiency is most sensitive to changes in residual saturation, porosity and hydraulic conductivity of the fine facies, parameters that control how much water fine-grained sediments retain (residual saturation and porosity) and 550 how quickly they drain (hydraulic conductivity). The large uncertainty on these three metrics (Table 1) combine to produce large overall uncertainty on recharge efficiency (Fig. 8).

Across all three outcomes of interest, DGSA results show that parameters that describe fine-grained sediments are more influential than those that describe the coarse deposits (Fig. 10). The prior uncertainty of parameters that describe fine facies is often greater than those that describe coarse facies (Table 1), which may contribute to this difference in model sensitivity. 555 However, the difference in prior uncertainty between coarse and fine facies is not unique to our field site; in general, fine-grained sediments can exhibit a wider range of porosity and hydraulic conductivity values than coarse sediments (Freeze and Cherry, 1979). This phenomenon is exacerbated by the fact that fine-grained sediments are poorly characterized in well logs (Faunt et al., 2010) and by pumping tests (Neuman and Witherspoon, 1972). In addition, these results highlight an important nuance in recharge site evaluation; sites with a higher percentage of coarse deposits will typically have higher recharge efficiency, 560 but the properties of the fine-grained sediments control uncertainty on expected recharge efficiency at the site. Thus, future site evaluation efforts should focus on identifying sites with a high proportion of coarse-grained material and characterizing the fine-grained facies at those sites. More specifically, measuring the hydraulic conductivity, porosity and residual saturation (or antecedent water content) of fine-grained sediments will reduce uncertainty on projected recharge efficiency. Fine-grained facies with low porosity and high antecedent water content will not be able to trap as much recharge water, leading to higher 565 recharge efficiency.

DGSA results also show that none of the three outcomes of interest are sensitive to inundation frequency. Water managers use a complex list of criteria when allocating surface water for MAR (Alam et al., 2020); this result helps simplify that list by allaying concerns over applying inundation at an ideal recurrence interval. However, inundation frequency may impact other flood-MAR concerns not described here, such as nitrate leaching from the root zone (Murphy et al., 2021; Waterhouse et al., 570 2021). Model output is also insensitive to parameters describing spatial uncertainty, which implies that the line spacing of this tTEM survey is dense enough that interpolation between lines does not impact modeling results. In addition, the similarity in the standard deviation of coarse fraction (Fig. S5) between the orchard (6.8 m tTEM line spacing) and the adjacent field (25–40 m line spacing) implies that future surveys can use a coarser line spacing, which will speed up data acquisition and reduce costs.

## 575 5.3 Importance of appropriately defining recharge

The retention of water in fine-grained vadose zone sediments highlights the importance of choosing an appropriate metric for evaluating recharge sites. One popular statistic is the infiltration rate minus the evapotranspiration rate, an approximation of deep percolation. However, this metric counts increases in both vadose and saturated zone storage as recharge, even though





the goals of most MAR projects — limiting water table decline, preventing land subsidence and increasing saturated zone
storage (Morrell, 2014) — predominantly relate to the saturated zone. While water below the root zone will likely reach the
water table, it can take years to do so. The difference between the two metrics is significant; our simulation results show that
median deep percolation efficiency is 88.3% after 14 days, while median saturated zone recharge efficiency is only 32.7% after
two years. Because two sites could have similar deep percolation rates but different saturated zone recharge rates, future work
should strive to quantify saturated zone recharge when possible. We recognize, however, that calculating this metric at the field
scale can be difficult and that other recharge metrics are still valuable in some instances, given their simplicity.

## 6 Limitations

While modeling results are promising, we acknowledge several limitations of our approach. First, we generate all coarse
fraction realizations using a single inversion of the tTEM signal, even though the inversion process is nonunique. To account
for this uncertainty, future work could use multiple, equally probable inverted resistivity models (Figure 3a) as conditioning
data for sequential Gaussian simulation. Second, CPT locations were chosen so as to sample a wide range of resistivity values,
but all cone penetration tests hit refusal before reaching the water table. The lack of lithology data from the saturated zone
may represent additional uncertainty in the resistivity-lithology transform. However, the range of resistivity values observed
in overlapping CPT and tTEM profiles encompasses 97% of all resistivity values in the tTEM resistivity model, so we do
not believe this uncertainty is appreciable. Moreover, the large-scale aquifer structures in our coarse fraction realizations
agree with nearby well logs (Mid-Kaweah GSA, 2019) as well as with hydrostratigraphic sections derived from an airborne
electromagnetic survey line over the site (Knight et al., 2018).

In addition, model parameterization relies on the assumption that there are only two end member sediment types at the site
(coarse and fine) and that each cell is a mix of the two end members. In reality, CPT logs show heterogeneity within each
end member, especially among sediments classified as "fine-grained" (Fig. S2). We account for some of the uncertainty in
this assumption by varying end-member hydrologic parameters between each simulation. Future work could improve upon
our workflow by varying parameters within blocks of sediment classified as 100% fine- or 100% coarse-grained. Finally, our
implementation of falling head infiltration within ParFlow assumes that water is instantaneously and equally redistributed over
the orchard between one time step and the next. While this assumption is likely valid with larger ponding depths and greater
time steps, it may exaggerate the extent of preferential infiltration towards the end of inundation; in reality, surface roughness
and microtopography restrict the redistribution of surface water, especially as ponding depth decreases.

## 7 Conclusions

This communication explores recharge processes during managed aquifer recharge (MAR) through a highly heterogeneous
vadose zone typical of many alluvial sedimentary systems. We present a geostatistical and hydrologic modeling workflow for
evaluating three outcomes of interest — infiltration rate at the surface, residence time of water in the root zone and saturated





zone recharge efficiency — at MAR sites surveyed with geophysical data. We apply the workflow to an almond orchard in the southern Central Valley, California, USA, and use the results to identify key processes to consider when planning a MAR project. Based on our simulation results, we conclude the following:

1. Vadose zone heterogeneity strongly impacts recharge processes and should be considered when selecting a MAR site. Water infiltrates into the subsurface primarily through coarse-textured sediments, while fine-grained facies accommodate
a disproportionately large fraction of the long-term increase in saturated zone storage.

2. Within the vadose zone, capillary-driven flow traps water in fine-grained sediments following inundation. These sediments take years to drain, limiting recharge to the aquifer. Two years after inundation, the saturation increase in fine-grained sediments below the root zone and above the water table — where water is unavailable to either plants or pumping wells — accounts for an average of 37% of the water applied at the surface. Importantly, this process occurs in
three dimensions, implying that even if inundation is limited to coarse sediments, fine facies may still pull water away from coarse flowpaths.

3. Because vadose zone processes can limit recharge, care needs to be taken when selecting a metric for evaluating recharge sites. When the goals of a MAR project focus on changes in the saturated zone, researchers should evaluate sites using metrics that directly quantify changes in saturated zone storage.

4. Model output is more sensitive to changes in parameters that describe fine-grained facies than those that describe coarse facies. Fine-grained sediments are often poorly characterized in hydrogeologic studies; water managers can reduce uncertainty in forecasted MAR outcomes with improved measurements of the hydraulic conductivity, porosity and van Genuchten parameters of fine facies, as well as measurements of their antecedent water content.

5. Altering the frequency of inundation events over a 4-month span (e.g., a single 80-cm flood or sixteen 5-cm floods) did
not impact net recharge to the aquifer or the duration of root zone saturation. This result implies that water managers should inundate MAR sites whenever water is available, without worrying about an optimal recurrence interval for inundation. Other factors at a site, like the hydraulic conductivity of the fine-grained sediments, are stronger determinants of MAR outcomes.

*Code and data availability.*  All simulations were performed with ParFlow-CLM v3.10.0 (https://doi.org/10.5281/zenodo.6413322). Data
and model input files are available to the public on the U.S. DOE Environmental Systems Science Data Infrastructure for Virtual Ecosystem.

*Author contributions.*  Zach Perzan: Conceptualization, Methodology, Formal Analysis, Software, Visualization, Writing - original draft; Gordon Osterman: Data curation, Methodology, Software, Writing - review & editing; Kate Maher: Conceptualization, Methodology, Funding acquisition, Writing - review & editing



*Competing interests.* The authors declare that they have no conflict of interest.

*Acknowledgements.* This material is based upon work supported under the National Science Foundation Graduate Research Fellowship (Grant No. DGE-1656518) and a Stanford Graduate Fellowship, with additional support from the Stanford Woods Institute for the Environment (Realizing Environmental Innovation Program). We would like to thank Aaron Fukuda of the Tulare Irrigation District for input on the primary outcomes of interest, Meredith Goebel and Rosemary Knight of Stanford University for feedback on the resistivity-lithology transform, and Stanford University and the Stanford Research Computing Center for providing computational resources and support that
contributed to these research results.





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
