# Peer review of "Controls on flood managed aquifer recharge through a heterogeneous vadose zone: hydrologic modeling at a site characterized with surface geophysics"

_Hydrology and Earth System Sciences, 2022_

## Referee Comment (RC1)

Review of: Controls on managed aquifer recharge through a heterogeneous vadose zone: hydrologic modeling at a site characterized with surface geophysics, by Perzan et al

Summary and Recommendation

The paper describes a methodology for inferring coarse vs. fine grained fractions of the subsurface based on towed subsurface resistivity mapping (tTEM) calibrated to cone penetration test (CPT) data, in an almond orchard in the Central Valley California. Using various spatial and multi-variable statistics, this mapping is transformed to subsurface coarse-grained-fraction 3D realizations. Variably saturated flow model that simulates managed aquifer recharge by flooding part of the orchard in the spring is used to test hydrological (infiltration and recharge) and agricultural (root-zone saturation periods) farmers' interest. Results showed that coarse-grained structures accommodate the rapid recharge. Fine-grained sediments in the root zone are probable to cause long –term saturation affecting yield. Fine grained-blocks in the deeper unsaturated zone may retain significant volumes of the MAR operation water for years after the flooding. Infiltration, recharge and root-zone saturation were found sensitive only to the fine-grained fraction's hydraulic properties, therefore the authors conclude that better characterization of the fine-grain end member is needed to reduce uncertainty in similar Agricultural-MAR operations.

The innovative technical procedure including non-invasive and invasive geophysical methods, geostatistical, Monte-Carlo and multivariate statistics, and 3D variably saturated flow simulations, is of new and advanced nature and very well described, therefore a good fit for publication in HESS. I have some arguments on the overall understanding of recharge under thick unsaturated zone and the practical conclusion for Ag. MAR operation which I would like the authors to discuss and perhaps rethink. Therefore, I recommend moderate revisions.

Major Comments

1) As the authors describe nicely recharge is correlated good with surface input (precipitation, irrigation, flood-MAR) due to the pressure wave that propagates fast in the unsaturated zone and push deep-unsaturated older water to the water-table (sometimes called by old groundwater hydrologists as "the train-car model"). Today we can deal with the unsaturated-zone's retention and avoid the simplistic recharge coefficients used in many groundwater models. Nevertheless, in the life-time of an almond-grove (20-50 years) the result of retention of ~ 1/3 of the MAR water in the deep unsaturated zone after 2 years, is no news and hardly important and shouldn't be highlighted as a main result in the abstract. Higher storage in the deep unsaturated-zone will increase recharge in 1 of the following years of high input either due to a rainy year or MAR operation. The authors should discus and maybe reconsider the important implication of their analysis in the scope of Ag–MAR in orchards.

   More-ever, recent 3D simulations in heterogeneous variably saturated medium concerning transient flow from a drywell showed that fine-grained layers at the bottom of a dry well contribute to faster downward flow in the unsaturated zone and faster recharge - see Russo et al., 2022, WRR**,** **https://doi.**org/10.1029/2021WR031881. This phenomena is due to the turn over in unsaturated hydraulic conductivity during drying, where at increasing negative pressure head, fine-grain medium becomes more permeable than course-grain.

2) Although used often, the term "deep vadose-zone" is awkward, as vadose comes from Latin meaning shallow. I suggest to use deep unsaturated zone for the domain between the bottom of root zone and the water table (especially if it is of tens of meters thick).

Specific comments

1) L71 see also Rudnik et al., 2022, WRR for use of stochastic approaches in MAR
2) L76-77 as discussed in major comment 1, in transient heterogeneous unsaturated flow fine-grained layers can increase flow in drying periods. Perhaps change "restrict flow" to impact flow.
3) L113-126,Why not define the recharge of a 40 m deep aquifer as the downward flux at 39 m depth and stay with fluxes:1) it is straightforward and simpler; 2) the saturated zone part of a model may include sources and sinks (pumping wells) or transient head boundary conditions which have impact on the lateral flow not related to the MAR operation. Discuss.
4) Figure 3, caption last row, change "hydraulic conductivity" to saturated hydraulic conductivity
5) L 209-210, perhaps better: Algebraically the resistivity of a tTEM cell is described by the harmonic mean of the fine and coarse grain resistivity's as:
6) L249, May be better hydraulic functions than "water retention curve" (to include the unsaturated hydraulic conductivity function as well as the retention curve).
7) L 360—370, only complete saturation? Or defined from some threshold of high saturation (e.g. 95% saturation)?
8) L414 – 415 "or from the particular …simulation."  not clear, explain or discard if not important for the sensitivity analysis description.
9) L 438 "0.15 +- 0.29" 0.29 standard deviation? define explicitly
10) L 461 discard "( 7 acres)"
11) Figure 6 caption. Is it a single flooding of 0.8 m, or another scheme? Should be said in caption.
12) Figure 7: 1) What drives the Flux recharge after 6 years when the water table is back to its pre-MAR level, perhaps not a consequence of MAR (relates to specific comment # 3); 2) right hand vertical axis title - typo recharge.
13) L494-495, this is a trivial result no need for numbers and statistics.
14) L 540 "especially within the vadose zone", where else than the unsaturated zone?
15) L616-621 – conclusion 2 – As said in major comments, the significance of this result in Ag.-MAR in almond groves is minor. 63% in 2 years for free or cheap water is no good? And the rest 37% are not lost forever they will recharge in the next rainy/MAR year (unless the pre-MAR unsaturated zone was in really low water contents)
16) L 629-634 conclusion 5 – Hard to belief that 1 flooding of 80 cm did not cause more saturation in rootzone than 16 inundations of 5 cm with 1 week between inundations. Check! A weekly 5 cm irrigation should not cause saturation of the entire root-zone unless very poor percolation in the soil.
17) L633 saturated hydraulic conductivity rather than "hydraulic conductivity"

---

## Author Comment (AC1)

We thank the reviewer for their detailed, timely and encouraging feedback. We have updated the manuscript based on the reviewer's suggestions.

The attached document contains a point-by-point response to each reviewer comment, with original comments in black and our response in blue. We have also attached updated versions of the manuscript and the supplement.

**Summary and Recommendation**

The paper describes a methodology for inferring coarse vs. fine grained fractions of the subsurface based on towed subsurface resistivity mapping (tTEM) calibrated to cone penetration test (CPT) data, in an almond orchard in the Central Valley California. Using various spatial and multi-variable statistics, this mapping is transformed to subsurface coarse-grained-fraction 3D realizations. Variably saturated flow model that simulates managed aquifer recharge by flooding part of the orchard in the spring is used to test hydrological (infiltration and recharge) and agricultural (root-zone saturation periods) farmers' interest. Results showed that coarse-grained structures accommodate the rapid recharge. Fine-grained sediments in the root zone are probable to cause long–term saturation affecting yield. Fine grained-blocks in the deeper unsaturated zone may retain significant volumes of the MAR operation water for years after the flooding. Infiltration, recharge and root-zone saturation were found sensitive only to the fine-grained fraction's hydraulic properties, therefore the authors conclude that better characterization of the fine-grain end member is needed to reduce uncertainty in similar Agricultural-MAR operations.

The innovative technical procedure including non-invasive and invasive geophysical methods, geostatistical, Monte-Carlo and multivariate statistics, and 3D variably saturated flow simulations, is of new and advanced nature and very well described, therefore a good fit for publication in HESS. I have some arguments on the overall understanding of recharge under thick unsaturated zone and the practical conclusion for AgMAR operation which I would like the authors to discuss and perhaps rethink. Therefore, I recommend moderate revisions (between minor and major).

**Major Comments**

1) As the authors describe nicely recharge is correlated good with surface input (precipitation, irrigation, flood-MAR) due to the pressure wave that propagates fast in the unsaturated zone and push deep-unsaturated older water to the water-table (sometimes called by old groundwater hydrologists as "the train-car model"). Today we can deal with the unsaturated-zone's retention and avoid the simplistic recharge coefficients used in many groundwater models. Nevertheless, in the life-time of an almond-grove (20-50 years) the result of retention of ~ 1/3 of the MAR water in the deep unsaturated zone after 2 years, is no news and hardly important and shouldn't be highlighted as a main result in the abstract. Higher storage in the deep unsaturated-zone will increase recharge in 1 of the following years of high input either due to a rainy year or MAR operation. The authors should discuss and maybe reconsider the important implication of their analysis in the scope of Ag–MAR in orchards.
To address this point, we have:

1) clarified that the timescale of recharge is an important consideration for water managers, if not for agricultural growers: "For example, in California, USA, the Sustainable Groundwater Management Act requires groundwater agencies to implement plans to mitigate groundwater overdraft and avoid undesirable consequences of overpumping by 2034 (Morrell, 2014). In addition, the negative effects of groundwater overdraft (e.g., subsidence) will continue to occur until the recharge

pressure pulse reaches the saturated zone. Accurately quantifying for saturated zone recharge rates will help water managers make an informed decision on whether to perform flood-MAR at a site or to use a different MAR technique with a more immediate impact, such as recharge through injection wells" (lines 589-594 in the updated manuscript without track changes).

2) explained that future work should investigate the long-term importance of this process at sites with repeated inundation over multiple years: "the extent of capillary trapping of recharge water in fine-grained sediments likely decreases with successive recharge events. Future work should strive to quantify the increase in saturated zone recharge efficiency with repeat recharge events over successive years" (lines 597-599).

More-ever, recent 3D simulations in heterogeneous variably saturated medium concerning transient flow from a drywell showed that fine-grained layers at the bottom of a dry well contribute to faster downward flow in the unsaturated zone and faster recharge - see Russo et al., 2022, WRR, https://doi.org/10.1029/2021WR031881. This phenomena is due to the turn over in unsaturated hydraulic conductivity during drying, where at increasing negative pressure head, fine-grain medium becomes more permeable than course-grain.

We have included this study in our discussion of capillary barriers in both the introduction (line 47) and the discussion (lines 520-522).

2) Although used often, the term "deep vadose-zone" is awkward, as vadose comes from Latin meaning shallow. I suggest to use deep unsaturated zone for the domain between the bottom of root zone and the water table (especially if it is of tens of meters thick).

We have replaced the phrase "deep vadose zone" with "thick unsaturated zone" in the abstract (line 3) to limit confusion, given the Latin etymology of the word "vadose".

**Specific comments**

1) L71 see also Rudnik et al., 2022, WRR for use of stochastic approaches in MAR

We have added this article to the examples referenced on line 71 in the revised manuscript.

2) L76-77 as discussed in major comment 1, in transient heterogeneous unsaturated flow fine-grained layers can increase flow in drying periods. Perhaps change "restrict flow" to impact flow.

We have changed "restrict flow" to "impact flow".

3) L113-126,Why not define the recharge of a 40 m deep aquifer as the downward flux at 39 m depth and stay with fluxes:1) it is straightforward and simpler; 2) the saturated zone part of a model may include sources and sinks (pumping wells) or transient head boundary conditions which have impact on the lateral flow not related to the MAR operation. Discuss.

While the technique the reviewer proposes is simple, it does not account for recharge due to mounding of the water table and would thus underestimate recharge rates. For example, if the water table is initially at 40 m and mounds up to 35 m, that water would not be counted as recharge until it moves below 39 m. This delay constitutes an underestimate in recharge rate. Moreover, in this case, we do not have any sources or sinks within the model domain that may impact lateral flow across each boundary.

4) Figure 3, caption last row, change "hydraulic conductivity" to saturated hydraulic conductivity

We have changed "hydraulic conductivity" to "saturated hydraulic conductivity".

5) L 209-210, perhaps better: Algebraically the resistivity of a tTEM cell is described by the harmonic mean of the fine and coarse grain resistivity's as:

We have updated this sentence to read: "Algebraically, the resistivity of a tTEM cell is described by the harmonic mean of the resistivity of the fine- and coarse-grained layers:" (line 210).

6) L249, May be better hydraulic functions than "water retention curve" (to include the unsaturated hydraulic conductivity function as well as the retention curve).
We have updated this sentence to clarify that the van Genuchten model is used to calculate both saturation and relative permeability.

7) L 360—370, only complete saturation? Or defined from some threshold of high saturation (e.g. 95% saturation)?
We have clarified that we use a 90% saturation threshold (section 3.4.1). Ganot and Dahlke (2021) estimated that a minimum volumetric air content of 0.09 $m^3/m^3$ is required to limit anoxia during Ag-MAR. However, since porosity varies both within and between simulations, for simplicity we use a constant saturation threshold instead.

8) L414 – 415 "or from the particular …simulation." not clear, explain or discard if not important for the sensitivity analysis description.
We have added an example to clarify this point: "For example, one stochastic realization of the model domain may have a higher percentage of fine-grained facies than another, limiting recharge" (line 414-415).

9) L 438 "0.15 +- 0.29" 0.29 standard deviation? define explicitly
We have defined this explicitly in the text (line 438).

10) L 461 discard "( 7 acres)"
We have removed this parenthetical in the updated manuscript.

11) Figure 6 caption. Is it a single flooding of 0.8 m, or another scheme? Should be said in caption.
We have clarified that this simulation received a single, 0.8 m inundation event.

12) Figure 7: 1) What drives the Flux recharge after 6 years when the water table is back to its pre-MAR level, perhaps not a consequence of MAR (relates to specific comment # 3); 2) right hand vertical axis title - typo recharge.
We have edited the caption to explain that "even after the water table returns to its pre-MAR level at ~6 years, water content in the vadose zone remains higher than in simulations without recharge. As this water percolates to the saturated zone, it contributes to flux recharge." We have also corrected the typo in the vertical axis.

13) L494-495, this is a trivial result no need for numbers and statistics.
The reviewer is correct that this result is intuitive to vadose zone hydrologists. The broader scientific community (along with water managers, consultants and other MAR practitioners) may be unfamiliar with vadose zone water retention processes, however, so we have chosen to keep these statistics in the manuscript.

14) L 540 "especially within the vadose zone", where else than the unsaturated zone?
We have edited this sentence to read: "However, over the long term, fine-grained sediments within the vadose zone absorb a large fraction of recharge water" (line 540).

15) L616-621 – conclusion 2 – As said in major comments, the significance of this result in Ag.-MAR in almond groves is minor. 63% in 2 years for free or cheap water is no good? And the rest 37% are not

lost forever they will recharge in the next rainy/MAR year (unless the pre-MAR unsaturated zone was in really low water contents)

As discussed in the major comments point 1 above, we have extended the discussion (section 5.3) to clarify the importance of the timescale over which saturated zone recharge occurs. See lines 589-594 and 597-599 in the updated manuscript.

16) L 629-634 conclusion 5 – Hard to belief that 1 flooding of 80 cm did not cause more saturation in rootzone than 16 inundations of 5 cm with 1 week between inundations. Check! A weekly 5 cm irrigation should not cause saturation of the entire root-zone unless very poor percolation in the soil.

We have edited this point to clarify that our results may be specific to this site. We have also edited section 5.2 to clarify that outcomes of interest are not sensitive to inundation frequency because "successive inundation events stack on top of one another and reach the water table as one cohesive pulse" (lines 566-567) and because "the large uncertainty on other parameters in the sensitivity analysis (Table 1) may dilute variability due to inundation frequency. For example, inundation frequency may impact root zone residence time, but uncertainty on other parameters (e.g., hydraulic conductivity of the fine-grained sediments) contribute more strongly to the observed variability in each outcome of interest at this site" (lines 571-574).

17) L633 saturated hydraulic conductivity rather than "hydraulic conductivity"

We have changed "hydraulic conductivity" to "saturated hydraulic conductivity".

**References:**

Ganot, Y. and Dahlke, H. E.: Natural and forced soil aeration during agricultural managed aquifer recharge, Vadose Zone Journal, 20, https://doi.org/10.1002/vzj2.20128, 2021.

---

## Author Comment (AC2)

We thank the reviewer for their detailed and thoughtful feedback. These contributions will help strengthen the paper and improve future work on this topic.

The attached document contains a point-by-point response to each reviewer comment, with original comments in black and the authors' response in blue. We have also attached updated versions of the manuscript and the supplement.

**Summary and Recommendation**

Perzan et al. manuscript deals with modeling managed aquifer recharge (MAR) under an almond orchard using geophysical data and Monte-Carlo simulations to deal with a relatively deep heterogeneous vadose zone. The manuscript is well written, the figures are excellent, and the scope is well suited for HESS. The workflow of parametrizing the model using geophysical data is thoroughly done, and I like the authors' approach to quantifying MAR recharge quantity and dynamics (comparing simulations with and without MAR). Unfortunately, the authors did not provide any validation of the simulation results by measurement of groundwater level and water content.

To give a wide perspective of their work, I think the authors tried to present this case study as representing a typical flood-MAR operation, that can include both agricultural MAR (Ag-MAR) and dedicated MAR facilities. This is not the case, in my opinion, and I suggest that the authors change the focus of the paper (including the title) to "Ag-MAR" or if they prefer "on-farm flooding". This is not just semantics: dedicated MAR facilities usually have a limited spreading area that overlies high conductivity subsurface layers, while in Ag-MAR the tradeoff is an "unlimited" spreading area that overlies less conductive sub-surface. The submitted manuscript exactly deals with the second case.

We have updated the title and introduction to clarify the focus of this study. Specifically, we have modified our definition of flood-MAR so it is in agreement with the California Dept. of Water Resources, which defines flood-MAR as a "management strategy that uses flood water resulting from, or in anticipation of, rainfall or snowmelt for groundwater recharge on agricultural lands and working landscapes, including but not limited to refuges, floodplains, and flood bypasses" (CA DWR, 2018; CA DWR, 2019). Our updated definition of flood-MAR no longer references dedicated recharge basins and clarifies that the scope of our study is limited to "agricultural land and working landscapes" (line 28 in the updated manuscript without tracked changes).

In addition, we believe that our methodology and results from the first and third metric (infiltration rate and recharge efficiency) will be of interest to flood-MAR practitioners beyond agricultural land, such as refuges and restored floodplains.

I have some concerns with the modeling of the infiltration process, calculation of the root zone residence time (discretization and root zone sizes are similar), and some suggestions for improving the manuscript (see details below). Overall, I recommend minor revisions.

**General comments**

I strongly suggest revising the Methods section. It is too long - about half of the manuscript. Although very informative, some parts of it should be moved to the supplementary material, mainly parts that were published previously (e.g., Fig. 4 and related text which is a similar approach to Goebel and Knight, 2021).

We have moved this figure to the supplement and edited section 3.2.3 to make our description more concise. Note that portions of our approach – specifically sections related to the saturated zone (lines 218-226) and our stochastic inversion process (lines 233-239) – have not been previously reported in the literature (in Goebel and Knight or elsewhere), so we cannot move this material to the supplement.

Please elaborate more on the impact of model discretization on the infiltration process. The vertical discretization of your model is very rough (1 m) for numerical infiltration problems where usually discretization is on the order of 0.05-0.3 m (e.g., Botros et al. 2012). Moreover, this may have a profound impact on your root zone residence time analysis – because your vertical discretization (1 m or more) is the same size as the root zone (1 m). Hence, the root zone in your model is probably only one cell that has no spatial dynamics – at each time step it always has only one value of water content. Please explain if this can impact your analysis (e.g., overestimating root zone residence time).

We have updated the supplement with additional simulations using a finer discretization scheme for the root zone. Specifically, we ranked all simulations by mean infiltration rate and randomly chose one simulation from each infiltration rate quartile (i.e., one with infiltration rate between 0-25th percentile, one from 25-50th, etc) to re-run with the upper meter of the domain discretized into 8 layers (0.05, 0.05, 0.15, 0.15, 0.15, 0.15, 0.15, and 0.15 m). Note that this discretization scheme causes a significant increase in computational time, so we are unable to use it across all simulations. Results from this analysis show that root zone saturation errors using our simplified (coarse) discretization scheme are <6% (Table S1). In addition, errors in root zone saturation are concentrated to coarse-grained model cells (Fig. S7). Given that coarse-grained cells rarely exceed the 48-hour root zone saturation threshold, this error does not alter maps of model cells that exceed the saturation threshold (i.e., the maps used to generate Fig. 4b).

We have also updated section 6 to note that the tTEM unit is unable to capture decimeter-scale heterogeneity within the root zone that could impact infiltration rates.

**Specific comments**

L76-77: This is the motivation of the study, but it is not convincing, as other studies explore "The influence of meter-scale heterogeneity on recharge.." – I suggest to mention some of these studies and their findings.

We have updated this sentence with citations to several studies that examine the impact of meter-scale heterogeneity on MAR processes.

L95: The authors chose the term flood-MAR to describe their framework, but it may be too general for their specific agricultural flood-MAR operation. Their choice of 2nd metric of root zone residence time demonstrates this point. I suggest using the term "Ag-MAR" (e.g., Ganot & Dahlke, 2021; Levintal et al. 2022) or "on-farm flooding" (Bachand et al. 2014) instead of flood-MAR for most places in the text.

As discussed above, we have updated the title to clarify that this study is focused on flood managed aquifer recharge. We have also removed any reference to dedicated recharge basins throughout the text, clarifying that the scope of our study is limited to "agricultural land and working landscapes" (line 28).

L111-112: In SAT-MAR anoxic conditions are one of the ways to reduce nitrate loads by denitrification.

We have added this point to the description of root zone residence time, clarifying that "Soil anoxia typically controls the removal of nitrogen species during soil aquifer treatment (Idelovitch and Michail, 1984)" (line 112).

L135-137: It is unclear why the authors chose for this work an agricultural field with a SAGBI rating of "poor" (maybe because of the available geophysical data?). This is an agricultural field that is not suitable for Ag-MAR according to the approach of O'Geen et al. (2015) which is much less exhaustive than the approach presented in the current paper. Please explain.

Correct. This site was chosen because of the availability of geophysical data. We have updated the site description to state this choice.

Table 1: Check the residual saturation (# 11 and 12). Values are too high (above common porosity values) – I believe it should be relative residual saturation.

We have updated this table to clarify that these are relative residual saturation values.

L319-321: Not sure how old is the almond orchard at your site, but during very long spin-up times (>15-25 years) other crops grown at the site with different Kc may have a different impact on water budget and subsurface storage. In other words, it seems like very long spin-up times (such as 131 years) have a little practical benefit.

Long spin-up times are required to initialize water content throughout the vadose zone. We have updated the methods section to note that "this value [0.15 m/yr] constitutes the best estimate of modern recharge rates, though it is of the same order of magnitude as the long-term average deep percolation rate across the entirety of the Central Valley (0.27 m/y; Faunt, 2009)" (lines 318-320). We also note that deep percolation rates in the Tulare region should be lower than the average rate reported by Faunt (2009), since the Tulare region has lower precipitation and higher evapotranspiration than the Central Valley as a whole.

L335-342: The falling head boundary is elegant, but probably far from real-life water application of a large plot with an area of 800 m x 400 m. This is mainly true for an impractical initial condition with a ponding head of 0.8 m (!). From my experience ponding during on-farm flooding rarely exceeds 10 cm. Unrealistic high ponding conditions may also overestimate infiltration rates. An alternative upper boundary will be to keep a constant head of 5 to 10 cm followed by a falling head (e.g., for a total application of 20 cm, with a 5 cm constant head, let 15 cm infiltrate under a constant head, and then start the falling head loop for the remaining 5 cm).

We have noted this limitation in section 6 (lines 621-625). We thank the reviewer for the suggested infiltration scheme, as it will only help to improve further work in this area.

L375: a suggestion – consider writing eq. (3) directly as the difference between MAR and no-MAR simulations (it is like plugging eq. (3) into eq. (4) and then Q0 and S0 are eliminated). Methodology-wise, it seems a more intuitive way to present the net recharge by MAR (and then you don't need eq. 4).

We have combined these two equations into a new equation 3 and we have made the description more concise so as to shorten the Methods section, as requested in the "General comments" above.

L438-400: Please explain how you got such high infiltration rates for a field that is rated as "poor" by the SAGBI index. These infiltration rates (even the 0.06 m/hr) will redefine the site's SAGBI rate as "good" or "excellent". Maybe the range of Ks that you used (#1 in Table 1) was too high? You disregarded half of the failed simulations because permeability was too low for the specified upper boundary flux (L432-434). It could be that the specific site cannot accommodate the amount of applied water, and therefore you preferred the converged simulations with high Ks. In other words, it could be that the Monte Carlo approach produced an overestimation of Ks and infiltration rates?

We have updated the site description with the caveat that U.S. Department of Agriculture Natural Resources Conservation Service (USDA-NRCS) soil maps – which SAGBI uses to derive deep percolation and root zone residence time estimates – historically only display 50% accuracy relative to high-resolution soil surveys (lines 140-142). In this case, this region was mapped in 2003 at the USDA-NRCS 1:24,000 resolution, which means that soil maps were first developed by analyzing aerial imagery, then a select few representative areas within the area were traversed to verify the imagery-derived delineations (Soil Survey Staff, 1993).

Saturated hydraulic conductivity estimates for each soil type constitute further uncertainty, as they were not directly measured during the survey. The $K_{sat}$ values were "based on soil characteristics observed in the field, particularly structure, porosity, and texture" (Wasner and Arroues, 2003, p. 184). Thus, the differences between this site's SAGBI rating and our findings may be due to imprecise maps of soil type and soil hydraulic conductivity.

Finally, we wish to acknowledge these caveats without detracting from the work of O'Geen et al. (2015); eight years later, this seminal study remains the most extensive analysis of MAR suitability across California.

L453: I suggest to rephrase "..accommodate more infiltration.." (maybe to "these cells have higher infiltration capacity..")
We have updated this line to read: "These cells have higher infiltration capacity because…" (line 453).

L462-464: Interesting result. I suggest emphasizing that your spring flooding is probably a riskier approach, as most growers will prefer to apply winter flooding when almonds (or other perennial crops) are dormant.
We have added this point to our discussion of inundation timing: "springtime inundation may introduce increased crop risk, as recharge operations may overlap with the growing season" (line 326-327).

L469: Fig. 6 and S8 are very nice. If doable, I suggest incorporating them both as one figure in the main text.
We have made the change as the reviewer suggests and combined these figures into a single Fig. 6.

L497-502: The dichotomic division to fine and coarse facies is arbitrary and subjective. Looking at the continuous parameters of the hydraulic functions, it is clear that all outcomes (obj. 1-3) are most sensitive to hydraulic conductivity, porosity, and residual saturation. So why divide the global sensitivity analysis into different facies?
Global sensitivity analysis quantifies model sensitivity to each input parameter parameter. Because values for coarse-grained and fine-grained end members are distinct input parameters (see section 3.3.2), we treat them separately within global sensitivity analyses.

L566-568: I would be cautious with declaring that the 3 outcomes are not sensitive to flooding frequency (especially root zone residence time). I think the source of this conclusion is site-specific – in your case infiltration rates are relatively high compared to the applied water volume. Inundation frequency of several wetting-drying cycles is common practice in MAR operations to maintain high infiltration rates (as you also stated in section 4.2.1). In SAT operations (and probably also in Ag-MAR) these cycles have also an important role in soil re-aeration.
We have edited this point to clarify that results may be specific to this site. We have also edited section 5.2 to clarify that outcomes of interest are not sensitive to inundation frequency because

"successive inundation events stack on top of one another and reach the water table as one cohesive pulse" (lines 566-567) and because "the large uncertainty on other parameters in the sensitivity analysis (Table 1) may dilute variability due to inundation frequency. For example, inundation frequency may impact root zone residence time, but uncertainty on other parameters (e.g., hydraulic conductivity of the fine-grained sediments) contribute more strongly to the observed variability in each outcome of interest at this site" (lines 571-574). In reference to the reviewer's point about soil aquifer treatment, we also note that "inundation frequency may impact other flood-MAR concerns not described here, such as nitrate leaching from the root zone" (lines 569-570).

**Technical corrections**

L59: change Harrington et al. (2014) to (Harrington et al., 2014)
We have made this correction in the manuscript.

L139: Consider changing the precipitation units to mm (60 mm and 450 mm)
We have updated the precipitation units in the manuscript.

L172: eq (1) – consider changing z to other letters, because you perform 1D vertical interpolation/extrapolation and in the following paragraph z denotes both resistivity [z(u)] and direction [z], which can be confusing.
To limit confusion, we have changed all references of "z direction" throughout the text to "depth direction". In this case, z(u) represents the z-score (or normal score) of resistivity. To maintain convention, we use z(u) to denote the z-score.

L319: consider changing "domain" with simulations or realizations (as you have only one model domain).
We have replaced "domain" with "simulation" in this sentence.

L426: Fig. S5 – while found in the SM, the axis title in Fig. S5c should be corrected to z(m); and the x=200 in the caption should be corrected to y=200.
We thank the reviewer for this astute observation and have corrected the y-axis in Fig S5c to "Z (m)". However, the x-axis in this subplot corresponds to the x direction, so we have left that label unchanged.

L458: Fig. S7 in SM, correct the legend to longer than 48 h (red/orange) and less than 48 h (blue). Also correct caption – "(green)" to "(red)"
We have updated the legend and caption in this figure to clarify that the blue curve represents the prior distribution (ie, coarse fraction across all root zone cells) and the orange curve represents the distribution of coarse fraction values that exceeded the saturation threshold.

Throughout the text – I suggest replacing "pressure" with "head" or "hydraulic head" as you also present it in units of length (e.g., Fig. S8).
Where appropriate, we have updated "pressure" to "pressure head" throughout the manuscript. For example, see lines 121, 249, 311, the caption to Fig. 5 and labels within Fig. 5.

L493-495: either use average or median for both facies (currently average use for fine, and median for coarse).
We have corrected this error in the text. The value for fine-grained facies was mistakenly written as "on average" when it is actually the median.

Fig. 10: isn't obj. 1 title "Preferential infil." should be changed to "infiltration rate"?

We have changed the obj. 1 title to "infiltration rate" as the reviewer suggests.

**References:**

CA DWR: Flood-MAR: Using Flood Water for Managed Aquifer Recharge to Support Sustainable Water Resources, Tech. rep., California Department of Water Resources, Sacramento, CA, https://water.ca.gov/programs/all-programs/flood-mar, 2018b.

CA DWR: Flood-MAR research and data development plan., Tech. rep., California Department of Water Resources, Sacramento, CA, https://water.ca.gov/programs/all-programs/flood-mar, 2019.

Soil Survey Staff, Soil Survey Manual, U.S. Department of Agriculture Handbook No. 18, Washington, DC, 1993.

Wasner, K., and Arroues, K., Soil Survey of Tulare County California, Western Part, Tech. Rep., National Resources Conservation Service, Sacramento, CA, 2003.